# Constructing synthetic nuclear architectures via transcriptional condensates in a DNA protonucleus

Miao Xie [1,2,3] ✉, Weixiang Chen [1,2,3], Maria Vonk-de Roy [1] & Andreas Walther [1,2] ✉

Nuclear biomolecular condensates are essential sub-compartments within the cell nucleus and play key roles in transcription and RNA processing. Bottom-up construction of nuclear architectures in synthetic settings is non-trivial but vital for understanding the mechanisms of condensates in real cellular systems. Here, we present a facile and versatile synthetic DNA protonucleus (PN) platform that facilitates localized transcription of branched RNA motifs with kissing loops (KLs) for subsequent condensation into complex condensate architectures. We identify salinity, monomer feeding, and KL-PN interactions as key parameters to control co-transcriptional condensation of these KLs into diverse artificial nuclear patterns, including single and multiple condensates, interface condensates, and biphasic condensates. Over time, KL transcripts co-condense with the PN matrix, with the final architecture determined by their interactions, which can be precisely modulated using a short DNA invader strand that outcompetes these interactions. Our findings deepen the understanding of RNA condensation in nuclear environments and provide strategies for designing functional nucleus-mimetic systems with precise architectural control.

In eukaryotic cells, the nucleus provides a compartment for essential processes such as transcription, mRNA pre-splicing, and ribosome assembly[1]. To ensure precise spatial and temporal regulation of these biochemical processes[2], membrane-less organelles such as nucleolus, Cajal bodies, and nuclear speckles form sub-compartments within the nucleus, which are biomolecular condensates that concentrate specific nucleic acids, enzymes, and metabolites[3–6]. Beyond regulating these crucial processes, unique nuclear patterns formed by biomolecular condensates vary across cell types, adapting to specific demands and functional cell states[7]. Importantly, dysfunctions in nuclear condensates have been implicated in diseases such as cancer, ribosomopathy, and neurodegeneration[6,8,9]. Thus, understanding and reconstructing nuclear biomolecular condensates is not only essential for uncovering their mechanisms but also holds significant potential for therapeutic applications.

Despite considerable advances in studying natural biomolecular condensates and attempts to engineer transcriptional condensates within the nucleus[8,10–12] based on specific or non-specific interactions of protein-protein, protein-nucleic acid, and RNA-RNA pairs[2,13,14], much still remains unknown about their formation mechanisms and the involved kinetic processes. Specifically, the mechanisms by which these condensates concentrate molecules, maintain structural integrity, regulate composition, and modulate internal biochemical activities remain elusive, largely due to the complexity of in vivo environments. In contrast, in vitro models of biomolecular condensates allow for precise control over composition in a simplified setting[11], enabling detailed mechanism assessment through experiments and computational modeling[15]. Here, studies presently however rely on plain solutions that are far from the conditions in a nucleus.

[1]Life-Like Materials and Systems, University of Mainz, Mainz, Germany. [2]Max Planck Institute for Polymer Research, Mainz, Germany. [3]These authors contributed equally: Miao Xie, Weixiang Chen. ✉e-mail: miao.xie@uni-mainz.de; andreas.walther@uni-mainz.de

Transcriptional RNAs with specific sequences have been identified to play a key role in many biomolecular condensation processes[15]. However, achieving control in synthetic nuclear architectures and functions requires more advanced RNA designs capable of forming higher-order structures. In nature, the self-complementary kissing loop sequence in type 1 human immunodeficiency virus (HIV-1) virions has been identified as framework for systematically manipulating genomic dimerization[16]. Similar kissing loop interactions have been shown to facilitate condensation in bacterial riboswitches[13,17]. Inspired by the sequence-dependent interaction of kissing loops, which enables specific pairing between internally folded RNAs[18,19], the groups of Takinoue[20], di Michele[21], and Franco[22] have recently introduced programmable condensates in solution formed by nanostar-like RNA motifs. The latter two groups have further shown that RNA nanostars with kissing loops at the end of each arm (KLs) could co-transcriptionally condense into condensates with controlled size, number, morphology, and composition either in solution or confined within water-in-oil emulsions[21,22]. Through integration of RNA aptamers into KLs, such condensates can mimic natural membrane-less organelles capable of selective capture of client molecules with biofunctions[21]. However, it remains unexplored whether RNA condensates can form in crowded conditions and how they may interact with DNA-rich environments resembling the cellular nucleus, where intricate RNA-DNA interactions occur. How such DNA environments influence the organizational principles of such designer condensates is unknown.

We have recently introduced core-shell DNA coacervates, formed by single-stranded DNA (ssDNA) polymers, with a highly concentrated DNA-enriched core[23], that can flexibly recruit molecules and proteins for enzymatic functions[24,25] and chemical reactions[26]. These DNA coacervates closely resemble the crowded environment of the cellular nucleus, making them an ideal platform for constructing nucleus mimics[27]. Therefore, we term them protonuclei (PN) in this study. As the internal composition of the PN can be flexibly tuned based on the ssDNA polymer selection, we incorporate T7 promoter sequences into the DNA core to recruit transcription templates and facilitate localized *in-protonucleo* transcription. We demonstrate that KL can be transcribed within these PN, leading to the formation of co-transcriptional KL condensates with various morphologies. We demonstrate a range of synthetic nuclear architectures, including single condensates, multiple condensates, interfacial condensates formed through secondary nucleation, and biphasic condensates of orthogonal KLs, all controlled by salinity, PN-KL affinity, and competing PN-KL interactions, respectively. Given the design flexibility of transcriptional KLs and the tunable condensate patterns in our crowded PN system, we believe this artificial nucleus platform will significantly advance the field of synthetic biology, in particular synthetic cells, providing a powerful toolkit for designing and constructing synthetic nuclear architectures with unprecedented control and precision.

## Results

Figure 1 shows an overview of our entire approach. It consists of constructing a modular PN platform using DNA nanoscience approaches, followed by immobilization of short KL templates to initiate transcription therein. The transcribed KLs are designed to undergo phase separation by complementary interactions. By precisely controlling KL-PN interactions and environmental conditions, we study structure formation and response in detail through easily accessible pathways. In more detail, the DNA PN are derived from our previous work on DNA protocells[23,24], where we have identified that temperature ramps of mixtures of long $poly(A_{20}-m)_n$ ssDNA and long $poly(T_{20}-k)_n$ ssDNA form micron-sized core-shell coacervates with an adenine-rich ssDNA polymer (polyA) core and a thymine-rich ssDNA polymer (polyT) shell[23–25,28]. This process features a selective liquid-liquid phase separation (LLPS) of polyA during heating, forming polyA droplets at high temperature, which are then stabilized by polyT with $A_{20}/T_{20}$ hybridization during cooling, forming a thin and crosslinked hydrogel shell. This ultimately furnishes a highly concentrated polyA core of around 10 g/L[29]. The dynamic properties of the PN can be regulated from an arrested state to a liquid-like state by tuning the salinity. Additional ssDNA barcode sequences (o, p, k) can be modularly incorporated into the ssDNA polymers for integrating functionalities into the core and the shell (Fig. 1).

We synthesized several ssDNA polymers using rolling circle amplification (details in Supplementary Table 1), including $poly(A_{20}-p)_n$, $poly(A_{20}-o)_n$, and $poly(T_{20}-k)_n$ with n ranging roughly from 10 to 60 repeating units[23]. The barcodes p, o, and k serve specific functions. The most critical part is the p barcode in $poly(A_{20}-p)_n$, which is the T7 RNA polymerase (T7 RNAP) promoter sequence that allows for the flexible integration of DNA templates (short genes) amenable to transcription of RNA in the PN through simple addition of the templates after formation of the PN. $Poly(A_{20}-o)_n$ serves to homogeneously dilute the p barcode and provides an addressable matrix barcode to tune properties and (as we will see below) adjust the affinity to the

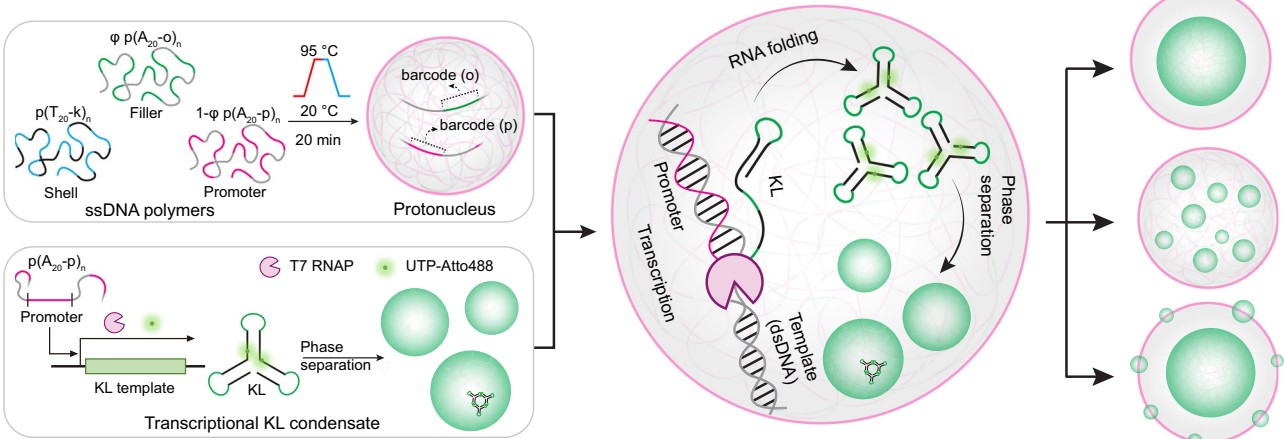

**Fig. 1 | Transcriptional kissing loop (KL) condensates form different synthetic nuclear patterns in DNA protonuclei (PN).** Adenine-rich ssDNA polymer (polyA) strands with barcode p (T7 RNA polymerase promoter sequence), polyA strands with dummy barcode (o), and thymine-rich ssDNA polymer (polyT) strands with barcode k are used for the LLPS process to form PN with an incorporated promoter region. The promoter barcodes inside the PN recruit DNA templates, T7 RNA Polymerase (T7 RNAP), and nucleotide triphosphate (NTP) monomers to induce a localized transcription and enrichment of KL sequences, forming distinct nuclear patterns via different nucleation and condensation processes.

transcribed RNA, which regulates the subsequent growth of the transcriptional condensates. Following our established protocols[23–25,28], we prepared a set of core-shell PN by mixing poly($A_{20}$-p)$_n$ and poly($A_{20}$-o)$_n$ for the core, and poly($T_{20}$-k)$_n$ for the shell, using a temperature ramp in TE buffer at 50 mM $Mg^{2+}$. Functionalization of the p and o barcodes with complementary dye-appended ssDNA confirms a homogeneous integration of both polyAs in the PN cores (confocal laser scanning microscopy (CLSM) images in Supplementary Fig. 1). The PN can be conditioned to different $Mg^{2+}$ concentration after preparation. We focus on a PN system where 10% of promoter sequences (poly($A_{20}$-p)$_n$) are diluted with 90% of a matrix (poly($A_{20}$-o)$_n$).

To verify transcription to occur inside the PN, we hybridized a transcription template $T_{x^*}$ (ssDNA template) containing p* for hybridization with the promoter sequence p and an active transcription region x* at stochiometric ratio into the PN (sequences in Supplementary Table 2). x* codes for a simple RNA not amenable to undergo condensation. Subsequent addition of T7 RNAP and a nucleotide triphosphate (NTP) monomer mix containing 1% fluorescent monomer (UTP-Atto488) induces transcription with local formation of fluorescent RNA strands (Fig. 2a, b, and Supplementary Movie 1).

To better quantify the transcription efficiency and kinetics inside the PN and compare it to free transcription in solution, we further designed a reporter (R) containing a fluorophore-quencher pair, which is a partially complementary double-stranded DNA (dsDNA; Rep/Rep' sequences in Supplementary Table 2;' denotes a partially complementary sequence leaving a toehold on Rep). The transcribed Rep* from template $T_{Rep^*}$ will trigger a strand displacement reaction (SDR) with R by fully hybridizing with the Rep strand, generating a fluorescent signal, which can be monitored by fluorescence measurements using a plate reader. In more detail, we compared the transcription kinetics among PN with embedded promoter sequence of poly($A_{20}$-p)$_n$, pure poly($A_{20}$-p)$_n$ ssDNA in solution, and short p ssDNA in solution —all at identical p concentration and under otherwise identical transcription conditions (Fig. 2a, c). All systems show relatively similar kinetic profiles, with the free promoter in solution being the most active transcription system, and the PN showing a slightly lower activity compared to the free poly($A_{20}$-p)$_n$ in solution. The slightly lower activity can be understood considering constraints of the diffusion of T7 RNAP and NTPs into the PN, and RNA strands out of the PN.

After confirming successful transcription in the PN, we turn to KL-condensate formation by transcriptional control in the PN versus in solution. As a proof of concept, we first focus on a three-armed singled-stranded RNA (ssRNA) nanostar with a wildtype palindromic KL sequence[20–22] at the tip of each arm (KL1 in Fig. 2d). We used a dsDNA template ($T_{KL1}$/$T_{KL1}'$) because ssDNA templates ($T_{KL1}$) alone do not allow for efficient transcription and condensate formation on account of intramolecular folding of such ssDNA templates (Supplementary Fig. 2). $T_{KL1}$ contains a p* ssDNA sequence for hybridization to poly($A_{20}$-p)$_n$ inside the PN to initiate transcription (sequences in Supplementary Table 2). We compared differences in KL1 condensate formation at low [NTP] = 3.6 × [$T_{KL1}$] after 18 h transcription. [NTP] is defined as the maximum amount of RNA transcripts that can be produced per template (see Method for details). The PN system clearly shows a single KL1 condensate in every PN with an average diameter of approximately 4 μm for PN with an average diameter of around 6.7 μm (Fig. 2e, f). In striking contrast, no KL1 condensates can be found in solution due to the limited concentration of RNA transcripts (Fig. 2e, f). Transcriptional KL1 condensates in solution start to appear with diameter of ~ 4.5 μm at increased [NTP] ([NTP] = 14.4 ×; Fig. 2e, f). The size of the transcriptional KL1 condensate in solution increases with [NTP] due to the increased amount of RNA transcripts (Supplementary Fig. 3). This comparison demonstrates that the spatial transcription of the KL1 in PN leads to locally high concentrations sufficient for condensation, similar to the enrichment mechanism in natural nuclear

condensates[2]. In addition, as we will demonstrate below, the KL-PN affinity also plays an important role in the condensation process.

Interestingly, one single KL1 condensate forms in each PN, confirming sufficient dynamics within the PN to follow energy minimization constraints to yield a minimum surface area (Fig. 2e and Supplementary Fig. 4). We further performed fluorescence recovery after photobleaching (FRAP) experiments on the KL1 condensates in PN and in solution to study their dynamic properties. Strikingly, their fluorescence recovery kinetics differ substantially. Whereas KL1 condensates in solution show near full recovery overnight, KL1 condensates in PN only show limited recovery, highlighting much better diffusion dynamics of individual KL1 RNA in KL1 condensate in solution than in PN (Fig. 2g, h). A complementary half-bleaching experiment shows a bright edge of transcriptional KL1 condensates in solution during recovery, indicating a dynamic exchange of soluble KL1 transcripts from the solution with the condensate phase (Supplementary Fig. 5). In contrast, half-bleached KL1 condensates in PN show less recovery and lack the bright edge, likely due to their restricted dynamics in a DNA-crowded environment and interactions between PN matrix and the KL1 transcripts, as we will further discuss below.

Next, we discuss the effects of [NTP] and [$Mg^{2+}$] on transcriptional KL1 condensate formation inside the PN. KL1 transcription with [NTP] varying from 2.4 × to 3.6 × show a morphological transition from peripheral localization of KL1 transcripts to reorganization and compaction into a single condensate (Fig. 2i). The formation of peripheral KL1 transcripts at low [NTP] shows that incoming NTPs are converted to RNA as they reach the embedded transcription templates in the outer PN parts. The lack of a centrally compacted condensate points to the fact that, at this low concentration of KL1 transcripts, phase segregation is at least not very pronounced. The remaining ring indicates an interaction between the PN matrix and the KL1 transcripts. At higher [NTP], KL1 transcripts are homogeneously produced throughout the PN, and phase segregation drives the formation of the KL1 condensate.

[$Mg^{2+}$] shows a profound impact on nucleation and condensate morphology. Multiple small condensates can be observed at 15 mM $Mg^{2+}$, whereas [$Mg^{2+}$] > 20 mM leads to the formation of a single condensate droplet. 20 mM $Mg^{2+}$ corresponds to a transition point. Interestingly, a transition in the condensate formation process is visible. Whereas isolated nucleation events dominate at 15 mM $Mg^{2+}$, co-continuous phase separation is visible above 25 mM $Mg^{2+}$ with a sponge-like structure. At 40 mM $Mg^{2+}$, condensate formation in PN is no longer visible (Fig. 2j). Such distinct condensate formation in PN is associated with multiple influences of $Mg^{2+}$ on the system: First, higher [$Mg^{2+}$] leads to reduced dynamics in the crowded environment of PN, as previously studied by us in detail[23,24,30]. Second, higher [$Mg^{2+}$] also assists in tighter condensation of the KL condensates and potentially increases non-specific interactions between the KL condensates and the PN matrix[22]. Third, increasing [$Mg^{2+}$] leads to a continuous decrease of the transcription kinetics as depicted in Fig. 2k. Thus, multiple isolated nucleation events and binodal phase separation occur at 15 mM $Mg^{2+}$, driven by the high dynamics of the PN core and the high transcription kinetics. In contrast, spinodal or viscoelastic phase separation[26,31,32] is favored at high $Mg^{2+}$ concentrations, where the dynamics of the PN become more arrested (see also discussion for time-lapse data at Fig. 3b–e). Further, charge screening increases the propensity for non-specific interactions between nucleic acids (RNA and DNA). The transcription is strongly suppressed at 40 mM $Mg^{2+}$ with limited KL1 transcripts so that condensation of KL1 cannot take place (Fig. 2j, k).

To get a deeper understanding of the morphological development of single KL condensates in the PN at 30 mM $Mg^{2+}$, we monitored the whole process over 24 h through CLSM (Fig. 3a, b). Two distinct stages occur. In the first 12 h, transcription takes place from the edge of the PN to their center due to the continuous consumption of NTPs as well as diffusive uptake of T7 RNAP and NTPs. The KL1 intensity

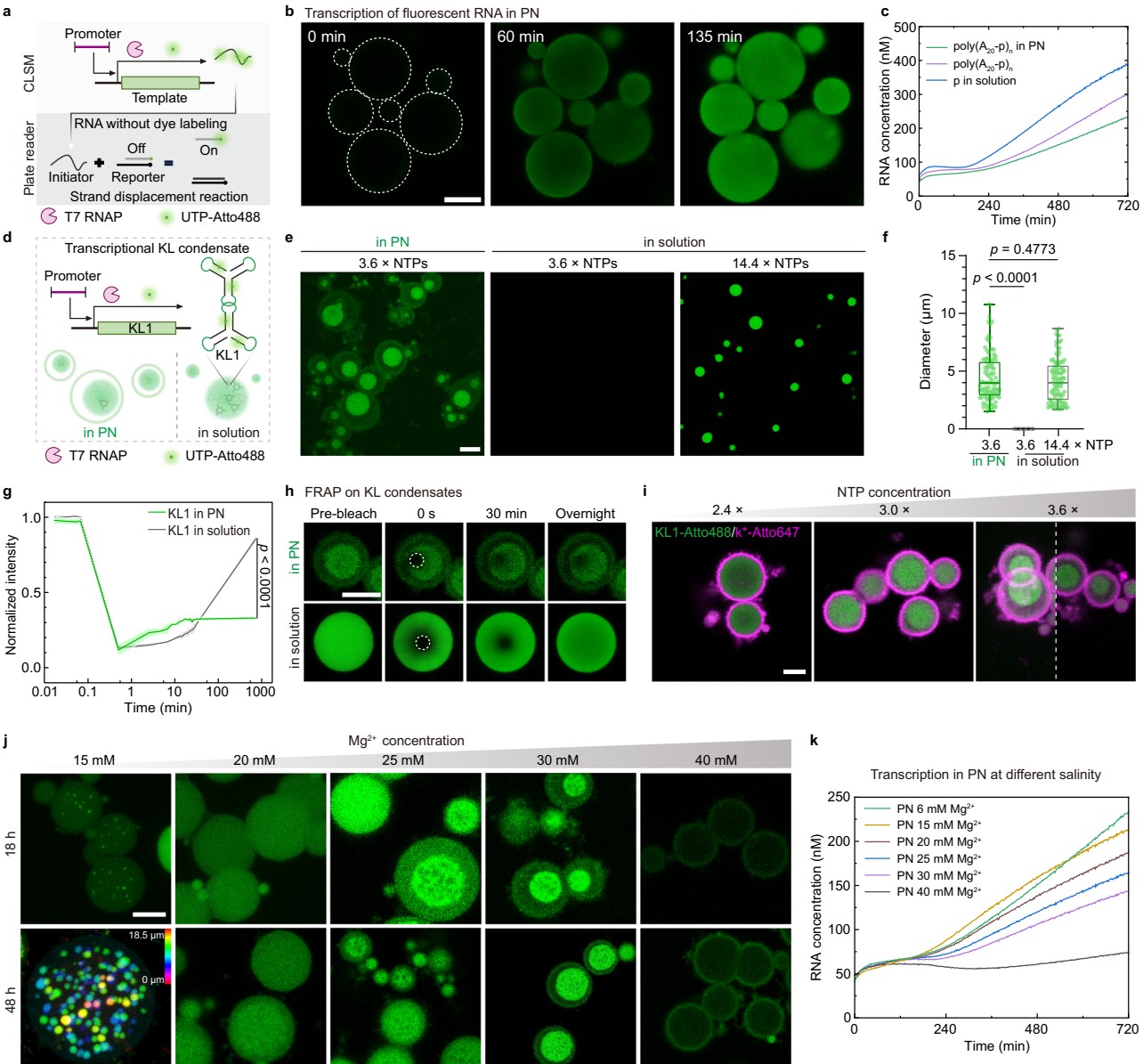

**Fig. 2 | Transcriptional KL condensates in PN show different nuclear patterns. a** Scheme of transcription in PN characterized by CLSM and plate reader. UTP-Atto488 labels transcripts for CLSM. In plate reader assays, dsDNA reporters (R) with fluorophore-quencher pairs generate fluorescence via RNA-triggered strand displacement reaction (SDR). **b** Representative CLSM showing localized transcription process of x* (green, UTP-Atto488 labeled) inside PN ([NTP]: [$T_{x*}$]: [p] = 2500: 1: 1; Supplementary Movie 1). **c** Transcription kinetics in PN with embedded poly($A_{20}$-p)$_n$, in solution with poly($A_{20}$-p)$_n$, or free promoter (p), monitored by SDR ([NTP]: [R]: [$T_{Rep*}$]: [p] = 2000: 10: 1: 1, 6 mM $Mg^{2+}$, mean of $N$ = 2). **d** Scheme of transcriptional KL1 condensate formation in PN and solution; KL1 labeled by UTP-Atto488. **e** CLSM images (maximum z-stack projections) showing transcriptional KL1 condensates in PN with poly($A_{20}$-p)$_n$ ([NTP]: [$T_{KL1}$] = 3.6), and in solution with free p ([NTP]: [$T_{KL1}$] = 3.6: 1 or 14.4: 1). **f** Diameter distributions of KL1 condensates in PN and solution at varying [NTP] ($N$ = 100 condensates from 3 experiments). Box plot: median (central line), interquartile range (box), min-max (whiskers). **g, h** FRAP on KL1 condensates in PN and solution at 30 °C. **g** Normalized fluorescence recovery kinetics in the bleached areas (**h**), with intensity normalized to pre-bleach levels (mean ± SD, $N$ = 3). **h** Time-lapse CLSM images with dashed circles marking bleached areas. **i** Representative CLSM images showing effect of [NTP] on KL1 condensate formation in PN. The left half of the 3.6 × [NTP] image is a z-stack projection. Green channel: KL1-Atto488; Magenta channel: k*-Atto647/ poly($T_{20}$-k)$_n$. **j** Representative CLSM showing effect of [$Mg^{2+}$] on transcriptional KL1 condensate formation in PN (15 mM $Mg^{2+}$ image is a color-coded z-depth projection). **k** Effect of [$Mg^{2+}$] on transcription kinetics in PN monitored via SDR (as in (**c**) with indicated [$Mg^{2+}$], mean of $N$ = 2). Two-sided $t$-test (**g, f**). Scale bars: 10 μm (**b, e**); 5 μm (**h, i, j**). Source data are provided as a Source Data file.

gradient can be diminished if the PN are pre-equilibrated with T7 RNAP (for 2 h) prior to the addition of the NTPs (Supplementary Fig. 6). The overall structure formation proceeds however in a very similar fashion. The entire structures reach maximum fluorescence intensities at 12 h (Fig. 3b–d). Spongy structures of KL1 condensates during phase separation start to appear at ca. 8–10 h, whereas significant coarsening and compaction into single spherical condensates follows in the later

12–24 h (Fig. 3b–e). The sponge-like, co-continuous structures of KL-PN co-condensates arises from spinodal or viscoelastic phase separation[26] of both polyA and KL transcripts in PN, that can appear in phase-segregating systems of high concentration of components and for situations of low dynamics. Different from the binodal phase separation via a nucleation and growth process for in-solution KL condensate formation at comparably high dilution[21], KL transcripts

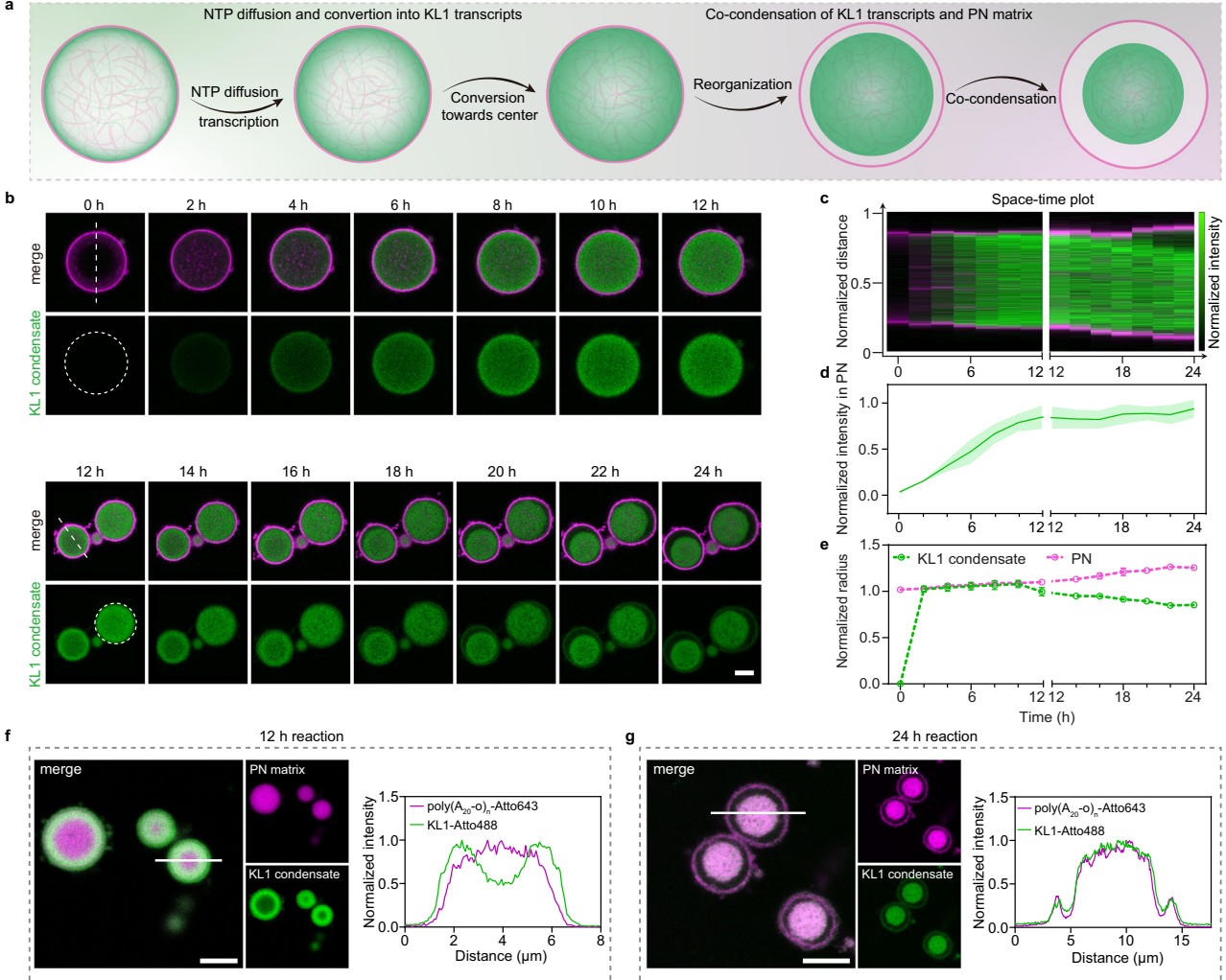

**Fig. 3 | Mechanism of co-condensation of transcriptional KL1 with the PN matrix. a** Scheme for formation of single co-condensates of transcriptional KL1 and polyA in PN through peripheral initiation of transcription, re-organization, and co-condensation. **b** Representative CLSM images of PN conducting KL1 transcription over 24 h ([NTP]: [$T_{KL1}$] = 3.6 : 1, 1 mol% UTP-Atto488). Note a slight slow-down of co-condensation kinetics of KL1 compared with results shown in Fig. 2e, j after 18 h, due to the interruption of shaking during incubation for CLSM imaging. Green channel: KL1 condensate labeled by UTP-Atto488; Magenta channel: PN shell (poly(T$_{20}$-k)$_n$ labeled by k*-Atto647). **c** Space−time plot analysis corresponding to the two dashed lines in (**b**) over 24 h shows the KL1 transcription, condensation, and reorganization process. **d** Fluorescence intensity in the KL1 condensate

channel over 24 h, measured within the two white dashed circles in (**b**). Intensity values were normalized to the 24 h intensity and expressed as mean ± SD from N = 3 PN measured. **e** Normalized radius of KL1 condensate and PN as measured from (**b**) over 24 h. Data are presented as mean ± SD (N = 3 PN measurements). Representative CLSM images of fluorescent PN (magenta) containing KL1 transcripts (green) at 12 h (**f**) and 24 h (**g**). The right plots correspond to line segment analysis of the white line in the CLSM images, showing fluorescence distribution for KL1-Atto488 and poly(A$_{20}$-o)$_n$-Atto643. The KL1 transcripts show a peripheral distribution in the PN matrix at 12 h (**f**), while colocalization and co-condensation between KL1 and PN matrix occur after 24 h (**g**). Scale bars: 5 μm (**b, f, g**). Source data are provided as a Source Data file.

transcribed inside PN continuously compact and co-phase-segregate with the PN matrix, ultimately forming interconnected networks. KL transcripts co-phase-segregating with the polyA matrix constitute slowly diffusing components with viscoelastic properties. Solvent molecules as the quickly diffusing component are expelled in the process, leading to interconnected network-like phase segregation that ultimately collapses into a spherical domain by interfacial energy minimization[33]. Interestingly, we can observe a relatively slow and continuous increase in the PN dimensions, as facilitated by the relaxation of polyA/polyT shell as a result of the increasing negative charge density inside the PN during localized RNA production and condensation (Fig. 3c, e).

For further probing the universality of this single condensate formation phenomena for various KL structures, we adapted a KL1 condensate with an RNA light-up broccoli aptamer (BrA) as one of the

arms, termed KL1-BrA (NUPACK-simulated structure shown in Supplementary Fig. 7a). After 12−24 h transcription, single condensates are formed in each PN. In contrast, KL1-BrA only forms irregular aggregates in solution. Here, the interaction between the PN matrix and the KL1-BrA condensate may facilitate better relaxation and stabilization of KL1-BrA condensate within the PN (Supplementary Fig. 7). Taken together, KL condensation in solution and in-PN differ profoundly in both the kinetic formation process and the formed final structures at what could be considered closer to the thermodynamic equilibrium. The system can be easily tuned by adjusting the NTP and Mg$^{2+}$ concentrations and is robust to changes in the KL components.

To study the behavior and aforementioned interactions of the PN DNA matrix with the KL1 RNA condensates, we covalently labeled poly(A$_{20}$-o)$_n$ with Atto643 to prepare fluorescent PN and used these new PN to initiate localized KL1 transcription. As expected, the initial

production and localization of KL1 transcripts occur at the periphery of the PN (Fig. 3f). Unexpectedly, co-condensation of the PN matrix with the KL1 condensates occurs over time. These co-condensates deposit at the bottom of PN after 24 h in the imaging chamber, highlighting their high density and compactness (Fig. 3g, Supplementary Fig. 8, and Supplementary Movie 2). FRAP experiments reveal a better recovery for the KL1 components compared to the PN matrix, corresponding to higher dynamics for the KL1 condensate composed of small RNAs than the PN matrix composed of long ssDNA polymers (Supplementary Fig. 9). This demonstrates the molecular level diffusivity of the RNA nanostars in this co-condensate structure.

Overall, this co-condensation between PN and KL1 condensate comes unexpectedly because the KL1 condensate is not designed to have any specific interactions with the PN matrix. Indeed, a NUPACK simulation suggests no specific hybridization between the $A_{20}$-o repeats and the KL1 sequence (Supplementary Fig. 10). Experimentally, we probed interactions between mature KL1 condensates and poly($A_{20}$-o)$_n$ inside PN by adding different quantities of o*-Atto647 (from 10 to 300%) that can bind to the majority phase of poly($A_{20}$-o)$_n$ in the PN. We hypothesized that the hybridization between o/o* may break non-specific KL1-PN interactions. To do so, we added different stoichiometric amounts of o*-Atto647 into solutions of PN containing already formed KL1-poly($A_{20}$-o)$_n$ co-condensates and investigated how the invasion by o*-Atto647 would alter the pre-existing KL1-poly($A_{20}$-o)$_n$ co-condensates. A gradual invasion of o*-Atto647 into the KL1-poly($A_{20}$-o)$_n$ co-condensates takes place as the amount of o*-Atto647 increases (Fig. 4a, b). A continuous surface erosion of the co-condensates occurs because the o/o*-Atto647 hybridization reduces the affinity between KL condensate and PN interior by introducing stronger electrostatic and steric repulsion inside PN due to increased negative charge density and increasing persistence length of the formed dsDNA parts[34,35] (Fig. 4c–e and Supplementary Movie 3). A sharp interface defined by a bright ring of o*-Atto647 with a locally high concentration appears[30]. The poly($A_{20}$-o)$_n$/o*-Atto647 thereafter occupies the space within the entire PN, whereas the KL1 transcripts are squeezed to the PN periphery and eventually dissolve into solution to equilibrate to their low concentration there. This process verifies that the interaction between KL1 and poly($A_{20}$-o)$_n$ PN matrix promotes the formation of the KL1-poly($A_{20}$-o)$_n$ co-condensate. Moreover, an invasion process of o*-Atto647 at 100 mM $Mg^{2+}$ shows a slower disassembly of the co-condensate compared to 30 mM $Mg^{2+}$ (Fig. 4c), implying the critical role of $Mg^{2+}$ in stabilizing interactions of KLs and KL-PN (Supplementary Fig. 11).

Seeing such a profound impact, we then investigated KL1 transcription in PN with a poly($A_{20}$-o)$_n$ matrix pre-hybridized by different amounts of o*-Atto647 (from 0% – 300%) to provide weakened affinity between PN matrix and KL1 transcripts. In analogy with the above result, single KL1 condensates form in pristine PN (Fig. 4f). When applying 10% o*-Atto647, the KL1 transcripts form single condensates with irregular secondary nucleation on its surface inside the PN, along with multiple tiny nuclei outside the PN shell (Fig. 4f). The brighter green parts are condensates purely enriched with KL1 transcripts that remain inside the PN due to relatively sufficient affinity. The marked difference – heterogeneously structured RNA condensates in Fig. 4f compared to the rather spherical and homogenous structures formed after invasion of pre-formed condensate in Fig. 4a, c – can be attributed to their different formation processes. Increasing the content of pre-hybridized o*-Atto647 domains from 10 to 300% gradually prevents KL1 condensate formation inside the PN due to weakened PN-KL1 interaction, which likely becomes even repulsive at higher pre-hybridization degrees because the ssDNA to dsDNA transition leads to higher negative charge density and persistence length, allowing for stronger electrostatic and steric repulsion[34,35], respectively, inside the PN. As a result, the KL1 transcripts formed inside the PN do not yield condensates insides the PN, instead, multiple small transcriptional KL1

condensates form in the PN surroundings. These results highlight the importance of the interaction between the DNA matrix of the PN and the KL1 transcripts in both the formation and the maintenance of the condensates within the PN. Hence, modulating the DNA-RNA interaction is a way for regulating nucleus condensate architectures.

To directly study the affinity between polyA sequence of the PN matrix and KL1 condensate, we prepared transcriptional KL1 condensates in solution and added $A_{20}$-o-Atto647 ssDNA, and $A_{20}$-o/o*-Atto565 dsDNA. Pure KL1 condensates sequester $A_{20}$-o-Atto647 whereas $A_{20}$-o/o*-Atto565 is excluded (Fig. 4g, h). Such marked differences among interactions between KL1-to-ssDNA versus KL1-to-dsDNA confirm some level of unspecific interaction between the KL1 transcript and the o region, which can be removed through hybridization into o/o*. Furthermore, electrostatic repulsion from increased negative charge density, and steric repulsion from higher persistence length after dsDNA formation also play important roles[34,35], as in analogy to re-entrant phenomena in living cells, where transcriptional condensate formation is promoted at low rates of RNA synthesis up to a point of charge imbalance, beyond which higher rates of RNA synthesis disfavors condensate formation[11,36].

Finally, we attempted to integrate orthogonal KL transcription systems into PN for constructing more complex structures to mimic multiple RNA condensates in the crowded environment of natural cell nuclei. We adapted two KL nanostars (KL1-R1 and KL2-R2) with orthogonal kissing loop sequences at the end of their arms, and distinct tail regions (R1 and R2) for specific labeling by R1*-Atto488 and R2*-Atto647, respectively (Fig. 5a). We firstly confirmed the transcription and the formation of centrally located condensates for both KL1-R1 or KL2-R2 inside PN (Supplementary Fig. 12a, b). Hence, both systems form similar condensate structure as the original KL1 system and the KL1-BrA system studied above.

Since [$Mg^{2+}$] can control the condensate morphology (Fig. 2j), we conducted co-transcription of both KLs in the same PN at 15 and 30 mM $Mg^{2+}$, respectively (Fig. 5a). At 30 mM $Mg^{2+}$, KL1-R1 assembles to a large single condensate (~0.7-fold the diameter of the host) at the PN center, while KL2-R2 forms small condensates, budding at the PN shell, with diameters less than 0.2-fold of the host PN (Fig. 5b–d). This suggests a preferred interaction between KL1-R1 and PN matrix, retaining the KL1-R1 condensate inside the PN, whereas KL2-R2 gets obviously expelled. KL1-R1 dominates the interaction with the PN matrix in this competitive system, whereas pure KL2-R2-PN would form a single central condensate (Supplementary Fig. 12b).

To ensure that this bias in structural development does not arise solely from different transcription efficiency, we conducted three control experiments: First, we compared the in-solution transcription efficiency of KL1-R1 and KL2-R2, finding out comparable transcription efficiency of both KL2-R2 and KL1-R1 (Supplementary Fig. 13). Second, we further screened whether biasing the in-PN transcription towards KL2-R2 by increasing its template concentration to a 3-fold excess over the KL1-R1 template would change the structure formation. However, even under such conditions, the co-condensate morphology does not invert, but KL1 condensates remain inside the PN and KL2 condensates form on the surface of the PN (Supplementary Fig. 14). Taken together, these results provide mutually reinforcing evidence supporting that the interaction between different KL and PN dominates the formation of the co-condensates and not the pure transcription efficiency. Third, we also probed whether KL2-R2 transcribed in solution can be enriched into pristine PN to check for interactions. Here, distinct differences are visible when comparing the structures formed by in-PN transcription of KL2-R2 versus KL2-R2 transcripts recruited from solution (Supplementary Fig. 12c). The in-PN transcription clearly induces phase segregation by KL2/KL2 interactions, whereas the latter rather points to some KL2-R2/PN interactions that enable a certain level of recruitment. Clear phase segregation via contraction of a spongy co-condensate phase is not visible for the latter. Overall, the

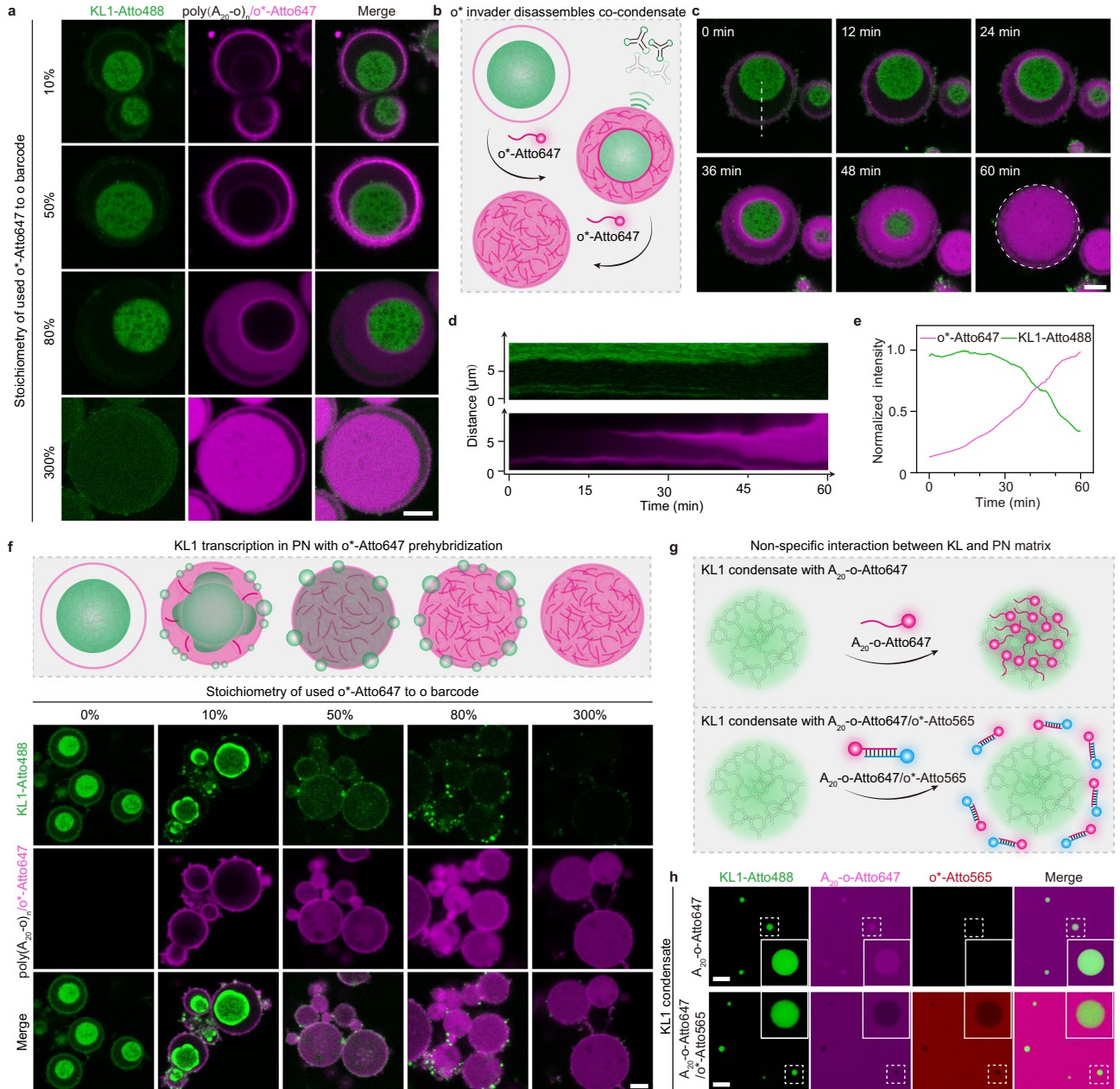

**Fig. 4 | Hybridization of the polyA matrix of PN induces disassembly of KL1-PN co-condensate. a** Representative CLSM images of transcriptional KL1 condensates (Atto488, green channel) in PN (Atto647, magenta channel) 60 min after adding 10%, 50%, 80% and 300% o*-Atto647 as an invader strand. **b** Schematic illustration of the o*-Atto647 invasion process. Hybridization of o*-Atto647 with poly($A_{20}$-o)$_n$ starts from the edge of the KL1-polyA co-condensate with a bright and sharp invading front to final dissolution of the whole co-condensate. **c** Representative CLSM images showing the process of co-condensate (Atto488, green channel) dissolution by adding 300% o*-Atto647 (magenta channel) to hybridize to the poly($A_{20}$-o)$_n$ of the PN. See also Supplementary Movie 3. **d** Space–time plot analysis along the white dashed line in (**c**) shows the gradual dissolution of the condensate. **e** Normalized fluorescence intensities of KL1 condensates (KL1-Atto488) and invader strand (o*-Atto647) measured in the white dashed circle in (**c**) during the

invasion process. Data are presented as mean values from $N = 2$ PN measured. **f** Scheme and representative CLSM images of KL1 transcription and condensation (Atto488, green channel) after 18 h in pre-hybridized PN with 0%, 10%, 50%, 80%, and 300% o*-Atto647 (magenta channel). **g** Scheme shows the attractive interaction between KL1 condensate and ssDNA $A_{20}$-o-Atto647, and repulsive interaction between KL1 condensate and dsDNA $A_{20}$-o-Atto647/o*-Atto565. **h** Representative CLSM images of pure transcriptional KL1 condensates (Atto488, green channel) prepared in solution, with the addition of (top) ssDNA $A_{20}$-o-Atto647 (magenta channel) for 1 h, showing preferential partitioning, or (bottom) dsDNA $A_{20}$-o-Atto647/o*-Atto565 (red channel) showing rejection. Three independent experiments show similar results for (**a, f, h**). Scale bars: 5 μm (**a, c, f**); 10 μm (**h**). Source data are provided as a Source Data file.

last control also emphasizes the critical role of in-PN transcription to facilitate co-condensate formation.

At 15 mM $Mg^{2+}$, the transcriptional KL1-R1 occupies the major PN space, while KL2-R2 forms multiple condensates in the PN (Fig. 5e–g). This can be attributed to weakened interactions between KL1-R1 and the PN at low salinity, allowing KL2-R2 to occupy some of the available

volume in the PN to form condensates. Note that a minor distribution of KL1-R1 in KL2-R2, or KL2-R2 in KL1-R1 (Fig. 5e, f), can be observed, possibly due to the slow diffusion of the produced KLs in the crowded condensates. Hence, the combined effect of $Mg^{2+}$ on changing the viscoelastic properties and modulating KL interactions as well as KL-to-PN interactions again shows a profound effect. We can conclude

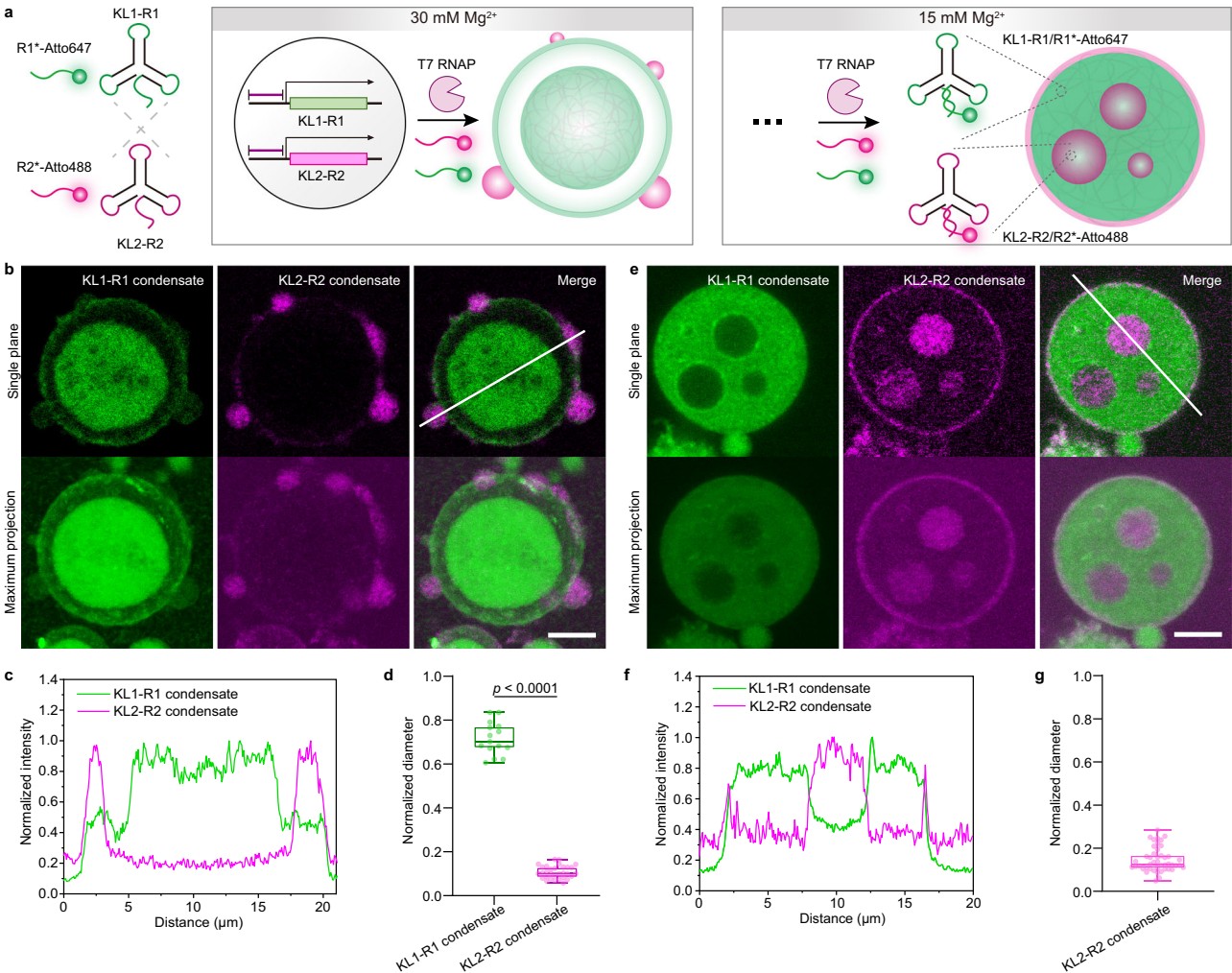

**Fig. 5 | Formation of orthogonal transcriptional KL condensates in PN.**
**a** Scheme showing orthogonal transcription and condensation of KL1-R1 and KL2-R2 in the same PN at different salinity. KL1-R1 is identical to KL1 in its nanostar framework, but with an additional recognition tail (R1) for R1*-Atto647 labeling. KL2-R2 shares the same stem sequence as KL1 but has orthogonal kissing loop sequences and a distinct recognition tail (R2) for R2*-Atto488 labeling. R1*-Atto647 and R2*-Atto488 are added during transcription. **b** Representative single-plane CLSM images and maximum intensity projection of z-stacked CLSM images showing orthogonal transcriptional condensates of KL1-R1 (green channel) and KL2-R2 (magenta channel) in PN at 30 mM Mg$^{2+}$ ([NTP]: [R1*]: [R2*]: [T$_{KL1-R1}$]: [T$_{KL2-R2}$]: [p] = 3.6: 1.8: 1.8: 0.5: 0.5: 1). **c** Normalized intensity profiles of line segment analyses corresponding to the white line in (**b**) for both channels. **d** Diameter of formed orthogonal condensates at 30 mM Mg$^{2+}$, normalized to the diameter of the

host PN. $N = 15$ for KL1-R1 condensate, $N = 31$ for KL2-R2 condensate, both from 3 independent experiments. **e** Representative single-plane CLSM images and maximum intensity projection of z-stacked CLSM images showing orthogonal transcriptional condensates of KL1-R1 and KL2-R2 in PN at 15 mM Mg$^{2+}$ ([NTP]: [R1*]: [R2*]: [T$_{KL1-R1}$]: [T$_{KL2-R2}$]: [p] = 3.6: 1.8: 1.8: 0.5: 0.5: 1). **f** Normalized intensity profiles of line segment analyses corresponding to the white line in (**e**) for both channels. **g** Diameter of formed KL2-R2 condensates at 15 mM Mg$^{2+}$, normalized to the diameter of host PN. Note that the diameter of KL1-R1 condensates cannot be quantified due to their hollow shape. $N = 46$ for KL2-R2 condensate, from 3 independent experiments. Box plot (**d**, **g**): median (central line), interquartile range (box), min-max (whiskers). Two-sided $t$-test (**d**). Scale bars: 5 μm (**b**, **e**). Source data are provided as a Source Data file.

that phase segregation of KL1-R1 is energetically favored to be retained in the PN. A mixing of both KL phases does not occur.

To verify the competitive interaction between KL1-R1 and KL2-R2 with the polyA in the PN matrix, we performed competitive partition experiments of A$_{20}$-o or A$_{20}$-o/o* with pure transcriptional KL1-R1 and KL2-R2 condensates grown in solution. The results show preferential partitioning of A$_{20}$-o into the KL1-R1 condensates, whereas A$_{20}$-o/o* is excluded by both condensates (Supplementary Fig. 15). This confirms a higher affinity of KL1-R1 condensates to the PN matrix and explains the different condensate architectures formed in the PN. NUPACK simulations further indicate that the origin of the preferred interaction between KL1-R1 and PN matrix is rooted in the tail structure (Supplementary Fig. 16).

Additionally, transcriptional KL2-R2 condensates grown in solution show a less spherical structure compared with transcriptional KL1-

R1 ones (Supplementary Fig. 15), suggesting stronger condensation interactions for KL2-R2 than KL1-R1, consistent with the higher melting temperature ($T_m$) of KL2 interactions than KL1 interactions provided in literature[21]. This also helps to explain the overall architecture of the multiphase structures in Fig. 5b, e from the perspective of surface tension ($\gamma$) of the ternary system of PN, KL1-R1, and KL2-R2[4,37]. Since KL2-R2 has a higher $T_m$ regarding the KL interactions, the binding strength among KL2-R2 in condensates is stronger and therefore KL2-R2 condensates possess higher surface tension than KL1-R1 condensates. In addition, KL1-R1 has a stronger binding interaction with the PN matrix than KL2-R2. This means $\gamma_{PN/KL2-R2} > \gamma_{PN/KL1-R1}$ applies, so that KL2-R2 condensate are preferably enveloped by KL1-R1 or expelled from PN to minimize its contact with the PN matrix. In summary, these results reveal that, in addition to salinity effects, subtle variations in RNA composition and sequence modulate their interaction with the

DNA matrix of PN in a competitive environment, leading to profoundly different condensation processes and resulting in distinct multi-phase co-condensate architectures in PN.

## Discussion

We have introduced a versatile nucleus-mimicking DNA condensate platform, a protonucleus, that enables localized transcription and the study of phase-separation of transcribed RNA nanostars in crowded and highly concentrated DNA environments. Since the strategy builds on our previous work on all-DNA synthetic cells[23–25,28], our approach shows how specific components from a completely different area of research, that is synthetic artificial cell research, can be effectively repurposed into new application domains. These PN offer a highly programmable platform for introducing short genes for transcription while also enabling control over properties such as gene density and the dynamic behavior of the matrix[23,24]. Transcription inside these crowded PN proceeds with satisfying efficiency up to high salt concentration. To study transcriptional folding and phase segregation in the crowded, nuclear-mimetic environment, we focused on KL condensates formed by ssRNA nanostars. We have identified that ionic strength is one key parameter for cross-regulating transcription efficiency, viscoelasticity of the PN, and KL-PN affinity. These effects in turn affect the nucleation of condensates from binodal to spinodal or viscoelastic phase separation[26,31,32], resulting in tunable artificial nuclear architectures inside the PN. The non-specific interactions between KL and PN matrix turns out to be crucial for retaining KL transcripts inside PN via KL-PN co-condensation. We show how such interactions can be efficiently modulated using DNA nanoscience approaches in such synthetic settings, ultimately leading to a repulsion and exclusion of the KL condensates from the PN.

We have further studied co-transcription and condensation of orthogonal KLs systems within the same PN, which results in distinct structures arising from competitive interactions between different RNA nanostars and the PN matrix. This highlights the potential of using our PN platform to study subtle interactions between RNA and DNA, as well as competitive interactions among RNAs in a DNA-enriched environment. Finally, at proper conditions, multiphase condensate structures can be built, which are further regulated by salinity through the cross-regulation of the viscoelastic environment, transcription efficiency, and competitive KL-PN interaction.

Looking into the future, our work opens new perspectives for constructing artificial nuclear architectures in synthetic model systems with DNA nanoscience tools. While we focus on a rather artificial and well controllable system of KL condensates, this work lays an important cornerstone to study more sophisticated phase separation processes, such as in case of polymerase II that forms rich condensate architectures with helper proteins, e.g., P-TEFb that are involved in transcription elongation, and those which are implicated in disease and ageing[36,38]. In addition, the modulable KL-PN interactions within protonucleus could serve as simplified models for transcriptional condensates in living cells, which are dynamically forming and dissolving, and essential for transcription regulations[11,36]. Moreover, from the perspectives of molecular systems engineering, synthetic biology, and artificial cell research, we have identified important pathways to transcriptionally regulate structure formation processes towards multiscale condensates that can be selectively addressed in their compartments. We anticipate that this system will serve as a valuable platform and toolkit for DNA nanoscience and synthetic biology.

## Methods

### Materials

ssDNA were purchased from Biomers and Integrated DNA Technologies (IDT). Supplementary Tables 1 and 2 summarize all sequences used in this study. T4 DNA Ligase (2 U/μL), Exonuclease I (40 U/μL), Exonuclease III (200 U/μL), and Φ$_{29}$ polymerase (10 U/μL) were purchased from Lucigen. Thermostable Inorganic Pyrophosphatase (2 U/μL), T7 polymerase (50,000 U/mL) and nuclease-free water were bought from New England BioLabs (NEB). Deoxynucleotide triphosphate (dATP, dTTP, dGTP and dCTP) (100 mM), Aminoallyl-dUTP-XX-ATTO-643 (1 mM), Aminoallyl-UTP-Atto488 (1 mM), and Aminoallyl-UTP-Atto630 (1 mM) were purchased from Jena Bioscience. Hexadecane, sodium chloride, magnesium chloride, Tris(hydroxymethyl)-aminomethane hydrochloride (Tris-HCl), Trizma base, acetic acid, and Ethylenediaminetetraacetic acid disodium salt dihydrate (EDTA), were purchased (as bioreagent grade if available) from Sigma-Aldrich. RNase Inhibitor (40 U/μL), RNase-free TE buffer (Invitrogen, 10 mM Tris and 1 mM EDTA, pH 8.0, 500 mL) were purchased from Thermo Fisher Scientific. 384-well high-content imaging glass bottom microplates were purchased from Corning.

### Instruments

All thermal annealing and heating ramps were performed on a TPersonal Thermocycler (Analytik Jena). Incubation with shaking was carried out on an Eppendorf ThermoMixer C with heated lid. DNA concentration was determined by a DS-11 Spectrophotometer (DeNovix). Confocal laser scanning microscopy (CLSM) was performed on a Leica Stellaris 5 (software: LAS X, v4.3.0.24308) with an oil-immersion ×63 objective.

### Synthesis of circular ssDNA templates and long ssDNA polymers

The synthesis of the circular DNA template and its corresponding ssDNA polymer are adapted from our previous reports[23]. In short, the linear ssDNA template and the corresponding ligation strand were firstly mixed at concentration of 1 μM in 100 μL TE buffer containing 100 mM NaCl. The solution was heated to 85 °C for 5 min before cooling to 25 °C with a cooling rate of 0.01 °C/s for hybridization. Afterwards, 20 μL of 10× Ligase buffer (500 mM Tris-HCl, 100 mM MgCl$_2$, 50 mM dithiothreitol and 10 mM ATP (Lucigen)), 70 μL of nuclease-free water, and 10 μL of T4 DNA Ligase (2 U/μL (Lucigen)) were introduced into the reaction mixture at room temperature for 3 h reaction. The solution was then heated to 70 °C for 20 min to deactivate the enzyme. Then, 10 μL of Exonuclease I (40 U/μL (Lucigen)) and 10 μL of Exonuclease III (200 U/μL (Lucigen)) were added into the reaction mixture for further overnight reaction at 37 °C to remove the ligation strands and any non-circularized templates in solution. Afterwards, the reaction mixture was heated to 80 °C for 40 min to deactivate the enzymes. To obtain the final circular ssDNA templates, the reaction mixture was washed with 400 μL TE buffer and filtrated using Amicon Ultra-centrifugal filters with a 10 kDa cut-off (Merck Millipore) for three times. The concentrations of the collected circular ssDNA templates were measured by the DS-11 Spectrophotometer (DeNovix), and the templates were stored in TE buffer at −20 °C.

For the synthesis of the long ssDNA polymers, we used rolling circle amplification (RCA). 5 μL of circular template (1 μM in TE buffer) and 1 μL of exonuclease resistant primer (10 μM in TE buffer) were mixed with 76 μL of nuclease-free water, 10 μL of commercial 10× polymerase buffer (500 mM Tris-HCl, 100 mM (NH$_4$)$_2$SO$_4$, 40 mM dithiothreitol, 100 mM MgCl$_2$ (Lucigen)), 2 μL of Φ$_{29}$ DNA polymerase (10 U/μL (Lucigen)), 1 μL of thermal stable inorganic pyrophosphatase (2 U/μL (NEB)) and 5 μL of adjusted deoxyribose nucleoside 5′-triphosphate mix (100 mM, the mix contains pure dATP, dTTP, dCTP, and dGTP solutions mixed in corresponding proportions of the exact composition of the desired ssDNA polymer repeating units (Jena Bioscience)). Note that for the synthesis of ssDNA polymers with in-chain fluorophores of Atto643, we replaced 2 mol% of the dTTP in the mix with Aminoallyl-dUTP-XX-ATTO-643 for random insertion of the dye into the ssDNA chains during RCA. The reaction mixture was kept at 30 °C for 50 h before thermal cleavage at 95 °C for 15 min to shorten the ultrahigh molecular weight of the synthesized DNA polymer[23]. The final products were purified by rinsing with 400 μL TE buffer and

filtration in Amicon Ultra-centrifugal filters with 30 kDa cut-off (Merck Millipore) three times. The concentrations of the collected final ssDNA polymers were measured using the DS-11 Spectrophotometer (DeNovix), and the DNA polymers were stored in TE buffer at −20 °C.

### Preparation of all-DNA PN embedded with T7 promoter sequence

The preparation of the PN is adapted from our previous reports[23] with modifications for the formation of PN containing T7 promoter sequence. Adenine-rich DNA polymers (poly($A_{20}$-p)$_n$ + poly($A_{20}$-o)$_n$ in a ratio of 1:9) (0.5556 g/L) and poly($T_{20}$-k)$_n$ (0.0694 g/L) were mixed in TE buffer without any salt at a final volume of 9 μL. The solution mixture was heated at 95 °C for 15 min for thermal cleavage to further reduce the chain length of the ssDNA polymers. Afterwards, 1 μL of TE buffer containing 500 mM MgCl$_2$ was introduced into the reaction mixture. The solution containing finally 0.5 g/L mixture of polyA and 0.0625 g/L poly($T_{20}$-k)$_n$ with 50 mM MgCl$_2$ was heated to 95 °C for 20 min (3 °C/s) and cooled down to room temperature (3 °C/s), yielding core-shell PN. Finally, the 10 μL of solution containing the PN was diluted 5 times by adding 40 μL of TE buffer containing various amounts of MgCl$_2$ to reach desired salinity. The obtained 50 μL of DNA condensates solution (as 5× diluted) has 0.1 g/L polyA mixture and 0.0125 g/L poly($T_{20}$-k)$_n$, corresponding to ca. 0.8 μM barcode p, ca. 7.2 μM barcode o and ca. 1 μM barcode k, respectively, in total solution. The solution was then stored in a fridge at 4 °C for 1 week for equilibration before usage.

### Spatially controlled transcription assay in PN

For transcription in PN monitored by plate reader, 3.125 μL of 5× diluted PN (90% barcode o + 10% barcode p) was further diluted into 25 μL of solution containing 1× RNA polymerase buffer (40 mM Tris-HCl, 6 mM MgCl$_2$, 1 mM DTT, 2 mM spermidine), 100 nM template ($T_{Rep*}$), 1 μM prehybridized fluorophore-quencher reporter (Rep/Rep' dsDNA), 2 U/μL T7 RNAP, 0.02 U/μL Thermostable Inorganic Pyrophosphatase, 1 U/μL RNase Inhibitor. MgCl$_2$ concentration was adjusted in different settings as noted in each figure caption. At the end, 2 μL of NTP mix (to reach 2 mM of ATP, GTP, CTP, and UTP each) was added into the solution to trigger the transcription reaction at different temperatures ranging from 25 to 30 °C. The final promoter sequence concentration in the solution is at 100 nM. As for control, transcription with pure promoter ssDNA (p) and poly($A_{20}$-p)$_n$ ssDNA polymer was also performed to compare the transcription efficiency.

For kinetic experiments under CLSM, $T_{x*}$ was loaded into the PN at a final concentration of 100 nM. Reporter was not used, instead, we further added 0.0833 mM Aminoallyl-UTP-Atto488 so that the transcribed RNA can be fluorescently labeled and observed under CLSM.

For each plate reader experiment, we have included a reference sample containing 1000 nM DNA-fluorophore conjugate. We calculate the intensity ratio between individual samples and the reference sample to yield the transcribed RNA concentration by following equation:

$$[\text{RNA}] = 1000 \text{ nM} \times \frac{\text{Intensity (sample)}}{\text{Intensity (reference)}} \qquad (1)$$

### [NTP] calculation

[NTP] is defined as the maximum amount of transcripts that can be produced per template given the nucleotide concentrations in the NTP mix in relation to the sequence of the transcript. For example for [NTP]: [template] = 5: 1, the [NTP] concentration is set in a way to at least allow for 5 full transcripts from 1 template based on the most abundant nucleotide in the transcript. Other NTPs will be in a slight excess as the NTP mix has equal stoichiometry for all four needed NTPs.

### Transcriptional KLs condensates formation

2.5 μL of 5× diluted PN (90% barcode o + 10% barcode p) was further diluted into 20 μL solution containing 1× RNA polymerase buffer (40 mM Tris-HCl, 6 mM MgCl$_2$, 1 mM DTT, 2 mM spermidine), 100 nM dsDNA template ($T_{KL1}/T_{KL1}'$, $T_{KL1-BrA}/T_{KL1-BrA}'$, $T_{KL1-R1}/T_{KL1-R1}'$, or $T_{KL2-R2}/T_{KL2-R2}'$; or 50 nM $T_{KL1-R1}/T_{KL1-R1}'$ + 50 nM $T_{KL2-R2}/T_{KL2-R2}'$), 2.5 U/μL T7 RNAP, 0.02 U/μL Thermostable Inorganic Pyrophosphatase, 1 U/μL RNase Inhibitor, and 0.048 mM NTP mix (0.048 mM of ATP, GTP, CTP, and UTP each at [NTP]: [$T_{KL1}$] = 3.6: 1, for maximum amount of KL1 produced, which is 3.6-fold of $T_{KL1}$, adjusted in different settings as noted in figure captions). MgCl$_2$ concentration was adjusted to 30 mM unless otherwise specified in figure captions. The mixture was incubated with shaking at 30 °C for 18 h reaction. The final promoter sequence concentration in the solution was at 100 nM. As control, transcription of KLs with pure promoter oligo was also performed with corresponding NTP concentration. For transcription of KLs with covalent label, 1 mol% of UTP was replaced by either Aminoallyl-UTP-Atto488, or Aminoallyl-UTP-Atto630 in transcription system. For KL1-BrA transcription, 0.05 mM DFHBI was added to the solution. For KL1-R1 or KL2-R2 transcription, 360 nM R1*-Atto647 or R2*-Atto488 sequence was added to the system, respectively. For transcriptional KL1 condensate formed in solution as a control, 100 nM promoter ssDNA was added to system, instead of PN.

### Fluorescence recovery after photobleaching (FRAP) experiments

FRAP experiments were performed by applying 3 times bleaching in a small circular region of interest (ROI) with diameter of 2 μm by 100% laser intensity. Post-bleaching images were recorded over different periods at 30 °C. The intensities within the circular ROI ($I_{ROI}$), and intensities in a circular region of the same size away from bleached region within the condensates ($I_{ref}$), in pre- and post-bleaching images were measured in ImageJ for performing double normalization in bleached regions by:

$$I_{\text{Norm}}(t) = \frac{I_{\text{ROI}}(t)}{I_{\text{ROI}}(t_0)} \times \frac{I_{\text{ref}}(t_0)}{I_{\text{ref}}(t)} \qquad (2)$$

to quantify the recovery kinetics over time. Note that $I(t_0)$ represents the intensity measured in the first image before bleaching.

### Statistics and reproducibility

The investigators were not blinded to allocation during experiments and outcome assessment. No statistical method was used to pre-determine sample size. No data were excluded from the analyses. Statistical parameters including the definitions and exact value of $N$ (e.g., total number of experiments, PN, or condensate), deviations, $p$ values, and the types of the statistical tests are reported in the figures and corresponding figure legends. Statistical analysis was carried out using GraphPad Prism 10.3.1 (509). Statistical analysis was conducted on data from three or more independent experimental replicates. Comparisons between groups were planned before statistical testing and target effect sizes were not predetermined. Error bars displayed on graphs represent the mean ± standard deviation (SD) or mean ± standard error of the mean (s.e.m.) from at least three independent experiments, as indicated in the corresponding figure legends. Statistical significance was analyzed using an unpaired Student's $t$-test for two groups. $p < 0.05$, $p < 0.01$, $p < 0.001$, and $p < 0.0001$ were considered significant. For Fig. 2c, k and Fig. 4e, mean values were presented from two independent experiments with similar results. All representative CLSM images were obtained from three independent experiments with similar results. CLSM images were analyzed by ImageJ (v2.9.0; 64-bit). Data analysis was performed on Microsoft 365 Excel (Version 2307) or MATLAB (R2023a). Origin 2023 (v10.0.0.154) or GraphPad Prism 10.3.1 (509) was used for plot.

## Reporting summary

Further information on research design is available in the Nature Portfolio Reporting Summary linked to this article.

## Data availability

Source data are provided with this paper. Additional supporting data are available from the corresponding author upon request. Source data are provided with this paper.

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

## Acknowledgements

We would like to thank Dr. Siyu Song and Tao Xu for their helpful discussion about data analysis and plotting. M.X. acknowledges the support of the Alexander von Humboldt Foundation. W.C. acknowledges support from the Max Planck Graduate Center with the Johannes Gutenberg University of Mainz (MPGC), and the RTG 2516 "Structure Formation of Soft Matter at Interfaces". This research was funded by the German Research Foundation (DFG) in the framework of the CRC 1551 "Polymer Concepts in Cellular Function" (R13). A.W. acknowledges funding from the Gutenberg Research Council Mainz underpinning his Life-Like Materials Program, the German Research Foundation grant WA 3084/19-1, the Max Planck Fellowship, and from the EU in the framework of the ERC Consolidator Grant to AW – M3ALI (101001638).

## Author contributions

M.X. and A.W. conceived the project. M.X. designed and performed experiments related to the system of transcriptional condensates in PN. W.C. developed the transcription system in PN. M.R. helped with the

initial transcription experiments. M.X. prepared the draft manuscript. M.X., W.C., and A.W. reviewed and edited the manuscript. A.W. supervised the project. M.X. and W.C. contributed equally.

## Funding

## Competing interests

The authors declare no competing interests.
