## [Peer Review File · Nature Communications]

Constructing synthetic nuclear architectures via transcriptional condensates in a DNA protonucleus

Corresponding Author: Professor Andreas Walther

Version 0:

Reviewer comments:

Reviewer #1

(Remarks to the Author)

This paper reports that nuclear-like DNA-based condensates that can transcribe RNAs. The DNA condensates were formed from three types of long single-stranded DNAs with a promoter sequence of T7 RNA polymerase. A transcribed RNA can fold into a three-branched structure with hairpin loops for a kissing-loop interaction. The transcribed RNAs were self-assembled into RNA condensates. This study revealed the behaviors and features of RNA condensates transcribed from DNA condensates, such as the localization of a formed RNA condensate in a DNA condensate. The experiments are well-organized, and the results are clear and interesting. This study is excellent research that will boost the DNA-based synthetic condensate field. The reviewer's questions and comments are below.

1) Figs. 2j, 3b, and 4a,c,f. What is the 'sponge-like' structure of RNA condensates? How did the structure form? Is it like bubbling caused by RNA degradation (doi.org/10.1073/pnas.2001654117)? Or is it a kind of phase separation (doi.org/10.1002/admi.202300898)?

2) Fig. 4: Why was the RNA condensate in Fig. 4f not spherical, different from the spherical RNA condensates in other results such as Fig. 4a and c? What decides the formed RNA condensate structures?

3) Fig. 5 and p.12, l.353-356 (This suggests a preferred interaction between KL1-R1 and PN matrix, retaining the KL1-R1 condensate inside the PN, whereas KL2-R2 gets obviously expelled. KL1-R1 dominates the interaction with the PN matrix in this competitive system, whereas pure KL2-R2-PN would form a single central condensate (Supplementary Fig. 9).): This phenomenon is very interesting. What decides the preference of the interaction between KL1-R1 and PN matrix and between KL2-R2 and PN matrix?

4) Fig. 5b: Why did the expelled KL2-R2 condensates attach on the outer surface of the PN, not inside the PN?

5) Fig. 5 and p.12, l.356-360 (At 15 mM Mg²⁺, the transcriptional KL1-R1 occupies the major PN space, while KL2-R2 forms multiple condensates in the PNs (Fig. 5e-g). This can be attributed to weakened interactions between KL1-R1 and the PN at low salinity, allowing KL2-R2 to occupy some of the available volume in the PN to form condensates.): Why did the stable structure (KL2-R2) have a weaker interaction with the PN matrix? What is the mechanism of this phenomenon?

6) Fig. 5e and p.12, l.356-360: Why did KL2-R2 form multiple condensates in the PNs? This behavior is different from KL1-R1. Can the authors explain this phenomenon based on the surface tension difference in both RNA condensates? Or is there another hypothesis?

7) Fig. 5: Are the asymmetric behaviors in the interaction of KL1-R1/KL2-R2 with PN related to not only the stability of KL1-

R1/KL2-R2 condensates but also the kinetics in the formation process of both the condensates?

8) Fig. 5: Does the asymmetric behaviors change if the ratio of the concentrations of the KL1-R1/KL2-R2 template DNAs in the PN?

9) Fig. 5: If only the KL1-R1 condensate is degraded after the formation of Fig. 5b structures, does the KL2-R2 condensate enter the PN and re-form a single condensate in the PN like Supplementary Fig. 9b?

Reviewer #2

(Remarks to the Author)

I have read with great interest the manuscript by Xie, Chen et al. The report combines the DNA based protocells developed by the Walther group with recently introduced RNA nanostructures capable of forming condensates upon in vitro transcription. The team uses these technologies to study co-transcriptional condensation in the crowded microenvironments of the DNA protocells that, arguably, shares many similarities with condensate formation in eukaryotic nuclei. The authors perform a wide array of experimental assays and controls and observe a number of intriguing effects including the key role played by interactions between the RNA nanostars and the surrounding DNA matrix in determining the characteristics (and presence) of the condensates.

Overall, I find this manuscript well written, and the results relevant to the interdisciplinary readership of the journal, particularly those interested in biomolecular condensates, synthetic cells and DNA/RNA nanotechnology. The document is well written and clear, and the figures are of high quality. I am happy to recommend publications after the authors have addressed the following minor comments:

1) In page 5, line 151, the authors introduce [NTP] as “defined as the maximum amount of KL1 transcripts that can be produced per template”. Could the authors clarify better this definition? Is this the overall NTP concentration divided by the number of nucleotides in one nanostar construct?

2) In page 5, line 158, the authors state “This comparison demonstrates that the spatial transcription of the KL1 in PNs leads to locally high concentrations sufficient for condensation, similar to the enrichment mechanism in natural nuclear condensates”, referring to the observation that under the relevant conditions KL1 condensates form in PNs but not in bulk. However, we find out later in the manuscript that the effect is likely due to affinity between the nanostars and poly(A20-o), so this statement is not really correct. Perhaps the authors could simply refer the reader to the later parts of the manuscript for a rationalisation of this effect?

3) Similarly, when discussing the difference in FRAP recovery between PNs and bulk (page 5, line 166), the authors could refer to the subsequent discussion on interactions between the KL1 nanostars and the DNA matrix, which is probably behind the slower recovery seen in PNs.

4) In Figs 2 c and k the authors convert a fluorescent signal from a molecular beacon into RNA concentrations. Could they please provide information on how this is done?

5) The authors use a single-stranded DNA template for transcribing the RNA nanostars. This will probably fold into a DNA nanostar similar to the RNA version, which may hinder transcription. Have the authors tested transcription using a double-stranded template? Does this change transcription efficiency in PNs or the bulk? Have they included GU wobble pairs in the RNA nanostar designs to destabilise the secondary structure of the single-stranded template?

6) Following up from 5: the authors ascribe the preferential condensation of KL1 stars within PNs to the affinity with the o domain in poly(A20-o), and they show good evidence for it. However, 10% of the scaffold is DNA in the PNs poly(A20-p), hybridised to single-stranded DNA templates that are likely to bin the transcription products (particularly when transcription ends and the polymerase loses activity). The authors could perhaps test or discuss this possibility?

Reviewer #3

(Remarks to the Author)

Within a core-shell DNA coacervates, the authors demonstrated that specific RNAs (kissing loop sequence, KLs) can be transcribed in-situ to form another phase of condensate inside, and this platform is termed as protonuclei (PNs). To construct this, they designed polyA and polyT containing ssDNA by incorporating the T7 promotor sequence or other barcode DNAs, and mixed with T7 DNA ligase. They provided the mechanism of co-condensation of transcriptional KL1 with the PN DNA matrix. This platform is capable of transcribing two KLs simultaneously that produces different patterns of multiphasic condensates by changes in magnesium concentration.

This work is original because the authors provided the first demonstration of in-situ co-transcription condensation KLs in PNs that can form multiphasic condensates. This work provides a clear observation of the KLs transcribed in PNs to

undergo condensation with DNA matrix. This work provides the new design strategy to construct the synthetic system that can perform the in-situ transcription with modality. This is potentially an interesting system for artificial cell research and synthetic biology.

While they provide thorough experimental designs to support their claims, there are a couple of remaining questions regarding their interpretation and observations:

1. If this core-shell DNA coacervates have a highly concentrated DNA-enriched core, why does the KLs transcription start from the inner-interface of PN shell (Fig 3b)? Is this a combined effect of NTPs diffusion and T7 polymerase from outside of DNA PNs? Perhaps, the T7 polymerase is enriched at the shell?
2. In this context, the authors describe the slower rate by T7 RNA polymerase in PNs caused by the slow diffusion of NTPs. Supposedly, this highly concentrated ssDNAs at the interface could suggest more dense networks of DNAs. Then, I would expect that T7 RNA polymerase diffusion would be slow as well, or even limited, because it is bigger than the NTPs. If this transcription is in a kinetic regime where the enzyme concentration doesn't affect the overall rate of reaction, the authors point of view is valid. Otherwise, some discussion regarding the T7 RNA polymerase encapsulation in PNs needs to be addressed.
3. Interpretation about the dsDNA exclusion from KL1 condensates in Figure 4 needs clarification.
 - a. If KL1 sequence does not have any specific interactions with the PN matrix, why would the hybridization of ssDNA with o* make KL excluded from the condensates? What is this interaction between KL1 and PN DNA matrix? Nonspecific interactions between KL and PN DNA matrix could be the charge interaction mediated by magnesium. Complementary sequence for PN DNA matrix would likely have partial dsDNA structure, which can yield in the increased charge density overall. If so, wouldn't it make these nonspecific interactions between KL and the polyA matrix stronger? Perhaps, magnesium is depleted to mediate the nonspecific charge interactions between PN matrix and KL1 that leads to the exclusion of the KL1? This could partially explain why the condensates inside of PNs become dominant by DNA matrix. I wonder higher magnesium concentration will make the co-condensates to be persisted upon adding o* invader.
 - b. Why does the KL1 condensate morphology by the prehybridization of DNA matrix in Fig 4f differ from Fig 4a? Is it because of different time point of imaging? Maybe the morphology in Fig 4a is kinetically arrested?
 - c. dsDNA can have greater persistence length that leads to exclusion from protein membraneless organelles in-vitro (Nott, T., Craggs, T. & Baldwin, A. Membraneless organelles can melt nucleic acid duplexes and act as biomolecular filters. *Nature Chem* 8, 569–575 (2016), some discussion about dsDNA vs ssDNA in Jeffrey R. Vieregge et al, *Journal of the American Chemical Society* 2018 140 (5), 1632-1638). I wonder whether this dsDNA material properties affect its partitioning trends in Fig 4 g & h.
4. It is still unclear how these two different KLs localize at the different locations under different magnesium concentration (Fig 5).
 - a. KL2-R2 condensates still have little amount of KL1 in both cases of magnesium concentrations, suggesting that the coexistence of KL1-R1 condensate and KL2-R2 condensate is more than sum of each individual condensates. Likely, DNA matrix for PN is involved. Please address that.
 - b. The reasoning of different morphology of KL condensates based on its material properties seem valid. This is a minor question, but do KL2 get transcribed inside of PNs and get transported to be outside of shell? Or does the KL2 get transcribed outside of the DNA PNs at 30 mM Mg²⁺?
5. In line 428, what does it mean by helper proteins?

Additionally, the manuscript does not specify the number of independent replicates. Some of fluorescence intensity curves have shaded areas representing standard deviations, but it is unclear where this standard deviation is coming from. Clarifying this information is important to improve the transparency and reproducibility of the study. The details of methods seem enough.

Version 1:

Reviewer comments:

Reviewer #1

(Remarks to the Author)

The authors have carefully revised their manuscript in response to the reviewers' questions and comments. New experimental data have been added, making it scientifically very interesting. Since the revised paper is very well structured, the reviewer recommends its publication in *Nature Communications*.

Reviewer #2

(Remarks to the Author)

The authors have addressed my minor concerns and I am happy to recommend publication.

Reviewer #3

(Remarks to the Author)

I am very pleased with the revised manuscript and appreciate its comprehensiveness and the authors' efforts in addressing the kinetics of RNA polymerase, diffusion of molecules, and condensation dynamics. I strongly recommend this manuscript for publication in *Nature Communications*.

We submit the revised version of our manuscript titled “Constructing synthetic nuclear architectures via transcriptional condensates in a DNA protonucleus”.

We thank the reviewers for their positive and constructive comments on the manuscript, and we appreciate the opportunity to discuss the reviewers’ comments and revise our manuscript accordingly.

We have considered all the comments and added requisite experiments and explanations to underscore specific points raised by the reviewers.

Summary of major additional experiments:

- Simulated interactions between orthogonal KOs transcripts and PN matrix by NUPACK to deepen the understanding on biphasic transcriptional condensates in PN.
- Effect of ssDNA or dsDNA templates on the formation of transcriptional KO1 condensates, both in solution and in PN.
- Effect of T7 RNAP diffusion on the transcription efficiency and the formation process of transcriptional KO1 condensate in PN by quantitative plate reader and visible CLSM.
- Confocal microscopy experiments demonstrate that increasing $[Mg^{2+}]$ from 30 mM to 100 mM slows down the disassembly of the co-condensate in PN during ω^* invasion.
- Investigation of impact of transcription kinetics on the asymmetric morphology of orthogonal KO1-R1 and KO2-R2 condensates formed in PN.
- CLSM experiment verifies that in-solution transcribed KO2-R2 could enter the pristine PN to some extent, whereas spongy co-condensate can only be formed by in-PN co-transcriptional condensation.

We trust that our revisions comprehensively address the points raised by the reviewers and the newly added experiments fall in line with expectations and broaden the scope of our study.

Reviewer #1 (Remarks to the Author):

This paper reports that nuclear-like DNA-based condensates that can transcribe RNAs. The DNA condensates were formed from three types of long single-stranded DNAs with a promoter sequence of T7 RNA polymerase. A transcribed RNA can fold into a three-branched structure with hairpin loops for a kissing-loop interaction. The transcribed RNAs were self-assembled into RNA condensates. This study revealed the behaviors and features of RNA condensates transcribed from DNA condensates, such as the localization of a formed RNA condensate in a DNA condensate. The experiments are well-organized, and the results are clear and interesting. This study is excellent research that will boost the DNA-based synthetic condensate field. The reviewer's questions and comments are below.

Thank you for your highly positive evaluation of our work.

1) Figs. 2j, 3b, and 4a,c,f. What is the 'sponge-like' structure of RNA condensates? How did the structure form? Is it like bubbling caused by RNA degradation (doi.org/10.1073/pnas.2001654117)? Or is it a kind of phase separation (doi.org/10.1002/admi.202300898)?

Response: Thank you for your insightful question. The sponge-like structure of KL-PN co-condensate originates either from spinodal or from viscoelastic phase separation that can appear in phase-segregating systems of high concentration of components and for situations of low dynamics, respectively. Different from the binodal phase separation via a nucleation and growth process for in-solution KL condensates formed at comparably high dilution¹, in-PN KL transcripts continuously compact and co-phase-segregate with PN matrix, ultimately forming interconnected networks, where the KL transcripts and polyA matrix, as slow-diffusion components, continuously expel the solvent molecule as the fast-diffusion component through spinodal or viscoelastic phase separation². Due to the distinct phase separation processes, the KL-PN co-condensate displays a rather rough surface (see single z-plane images in Fig. 2h) and loose structure (see the higher intensity of pure KL condensate on the outside PN shell and the lower intensity of co-condensate within PN in Fig. 4f), compared with pure KL condensates.

We add additional text to clarify this:

The sponge-like, co-continuous structures of KL-PN co-condensates arises from spinodal or viscoelastic phase separation³ of both polyA and KL transcripts in PN, that can appear in phase-segregating systems of high concentration of components and for situations of low dynamics. Different from the binodal phase separation via a nucleation and growth process for in-solution KL condensate formation at comparably high dilution¹, KL transcripts transcribed inside PN continuously compact and co-phase-segregate with the PN matrix, ultimately forming interconnected networks. KL transcripts co-phase-segregating with the polyA matrix constitute slowly diffusing components with viscoelastic properties. Solvent molecules as the quickly

diffusing component are expelled in the process, leading to interconnected network-like phase segregation that ultimately collapses into a spherical domain by interfacial energy minimization².

2) Fig. 4: Why was the RNA condensate in Fig. 4f not spherical, different from the spherical RNA condensates in other results such as Fig. 4a and c? What decides the formed RNA condensate structures?

Response: Thank you for your question. In Fig. 4a and c, RNA condensates are first formed in PN, which are **NOT** hybridized with o*-Atto647 (o*-Atto647 has the function to reduce affinity between KL condensates and the PN interior). o*-Atto647 is later added as an invader after RNA condensates were formed. That is why a spherical RNA condensate gradually dissolves from the outer surface to the center of PN.

In contrast, in Fig. 4f, KL transcription and condensation is conducted in PN **prehybridized** with different amounts of Atto647-o*. This hybridization weakens the KL-to-PN affinity, making RNA condensation within the PN difficult at 10% prehybridization and leading to ill-defined RNA condensates. One can also see the brighter edges of these condensates. These arise from rather pure KL condensates, whereas the less intense core is a co-condensate. The pure KL layer even appears to dewet from the central condensate as seen by the undulations/non-perfect bright shell. At higher prehybridization levels, the internal PN environment becomes more repulsive, which prevents RNA condensation inside PN. Therefore, small condensates form at the outer shell of PN due to diffusion of KL transcripts from the PN to the surrounding.

We find that the experimental description is insufficient in the original manuscript for showing the differences between experimental settings for Fig. 4a-c and Fig. 4f. Therefore, we now provide additional text and explanation for Fig. 4a-c as follows:

To do so, we added different stoichiometric amounts of o*-Atto647 into solutions of PN containing already formed KL1-poly(A₂₀-o)_n co-condensates and investigated how the invasion by o*-Atto647 would alter the pre-existing KL1-poly(A₂₀-o)_n co-condensates. A gradual invasion of o*-Atto647 into the KL1-poly(A₂₀-o)_n co-condensates takes place as the amount of o*-Atto647 increases (Fig. 4a, b). A continuous surface erosion of the co-condensates occurs because the o/o*-Atto647 hybridization reduces the affinity between KL condensate and PN interior by introducing stronger electrostatic and steric repulsion inside PN due to increased negative charge density and increasing persistence length of the formed dsDNA parts^{4,5} (Fig. 4c-e and Supplementary Video 3).

For Fig. 4f, we have also modified the original text and believe the relevant discussion (highlighted) now can stand alone and make a clear distinction to the discussion about Fig. 4a-c:

“Seeing such a profound impact, we then investigated KL1 transcription in PN with a poly(A₂₀-o)_n matrix pre-hybridized by different amounts of o*-Atto647 (from 0% – 300%) to provide weakened affinity between PN matrix and KL1 transcripts. In analogy with the above result, single KL1 condensates form in pristine PN (Fig. 4f). When applying 10% o*-Atto647, the KL1 transcripts form

single condensates with irregular secondary nucleation on its surface inside the PN, along with multiple tiny nuclei outside the PN shell (Fig. 4f). The brighter green parts are condensates purely enriched with KL1 transcripts that remain inside the PN due to relatively sufficient affinity. The marked difference – heterogeneously structured RNA condensates in Fig. 4f compared to the rather spherical and homogenous structures formed after invasion of pre-formed condensate in Fig. 4a, c – can be attributed to their different formation processes. Increasing the content of pre-hybridized o*-Atto647 domains from 10% to 300% gradually prevents KL1 condensate formation inside the PN due to weakened PN-KL1 interaction, which likely becomes even repulsive at higher pre-hybridization degrees because the ssDNA to dsDNA transition leads to higher negative charge density and persistence length, allowing for stronger electrostatic and steric repulsion^{4,5}, respectively, inside the PN. As a result, the KL1 transcripts formed inside the PN do not yield condensates inside the PN, instead, multiple small transcriptional KL1 condensates form in the PN surroundings.”

3) Fig. 5 and p.12, l.353-356 (This suggests a preferred interaction between KL1-R1 and PN matrix, retaining the KL1-R1 condensate inside the PN, whereas KL2-R2 gets obviously expelled. KL1-R1 dominates the interaction with the PN matrix in this competitive system, whereas pure KL2-R2-PN would form a single central condensate (Supplementary Fig. 9).): This phenomenon is very interesting. What decides the preference of the interaction between KL1-R1 and PN matrix and between KL2-R2 and PN matrix?

Response: Thank you for the question. In the original manuscript, we had experimentally verified the preferred affinity of KL1-R1/PN over KL2-R2/PN in original Supplementary Fig. 10 (now Supplementary Fig. 16 in revised manuscript). As KL1-R1 and KL2-R2 share the same stem sequence but have different tails (R1 and R2) for binding fluorescent labels, we hypothesize that the difference in the KL-PN affinity interaction might be rooted in the different tails. To further investigate this, we performed NUPACK simulations (rna06 model) at 30 °C with the following sequences (tail regions are underlined):

repeating unit of PN matrix p(A₂₀-o)_n (400 μM)

AAAAAAAAAAAAAAAAAAGCATTGAGGTATTGTTGCTCCCA

RNA sequence of KL1 (200 μM):

GGGAUGCACAUUAGAGUCGCUCAUCGCGAAAGAGCGGCUCUAGUGUGCUCGCGUGCCUCAGAGGAC
AUCGCGAAAGUCCUUUGAGGUACGCGUCACUCGUAGCAUUGUGCAUCGCGAAAGCACAGUGCUAUG
AGUGCAUCUGAACGAGUAAGGACCCCA

RNA sequence of KL2 (200 μM):

GGGAUGCACAUUAGAGUCGCUCAUCGCGAAAGAGCGGCUCUAGUGUGCUCGCGUGCCUCAGAGGAC
AGUCGACAAGUCCUUUGAGGUACGCGUCACUCGUAGCAUUGUGCAGUCGACAAGCACAGUGCUAUG
AGUGGGUGGCUUAUUUACAGGCGUUAG

As shown below, we find that at this condition, KL1-R1 indeed has a higher level of binding to the $p(A_{20-o})_n$ than KL2-R2. And the origin of this binding interaction comes from the tail of the KL structure, i.e., the R1 sequence. In contrast, KL2-R2 only shows a minor binding at a different position. We have added this new result to Supplementary Figure 16.

Supplementary Figure 16. Binding interaction between A_{20-o} , KL1-R1, and KL2-R2.

a, Simulated concentration distribution of complex sequences in a pool of A_{20-o} (400 μM), KL1-R1 (200 μM), and KL2-R2 (200 μM) at 30 $^{\circ}\text{C}$ based on rna06 model. Salt: 1 M Na^+ . **b**, Predicted minimum of free energy (MFE) proxy structure at 30 $^{\circ}\text{C}$ of A_{20-o} + KL1-R1 and A_{20-o} + KL2-R2. Tails of KL1-R1 and KL2-R2 are highlighted in orange. The binding between A_{20-o} and Kls are indicated by arrows.

We added additional discussion in the text:

NUPACK simulations further indicate that the origin of the preferred interaction between KL1-R1 and PN matrix is rooted in the tail structure (Supplementary Fig. 16).

4) Fig. 5b: Why did the expelled KL2-R2 condensates attach on the outer surface of the PN, not inside the PN?

Response: Thank you for your question. As KL1-R1 dominates the affinity interaction with the PN matrix, it co-condenses with the PN. The co-condensation between KL1-R1 and PN on one hand screens the affinity interaction between KL2-R2 and the PN matrix, and on the other hand induces electrostatic repulsion of KL2-R2 due to the highly dense and negatively charged properties. Therefore, the transcribed KL2-R2 will get expelled from the PN interior to the outside. Since KL2-R2 has a local high concentration just outside of the PN due to the in-PN transcription, KL2-R2 condensates form on the outer surface of the PN. We believe that this is now sufficiently and clearly explained in context with the new data provided (Updated Supplementary Fig. 12 and new Supplementary Fig. 16) and with the explanation existent in the text (see around page 13).

5) Fig. 5 and p.12, l.356-360 (At 15 mM Mg^{2+} , the transcriptional KL1-R1 occupies the major PN space, while KL2-R2 forms multiple condensates in the PN (Fig. 5e-g). This can be attributed to weakened interactions between KL1-R1 and the PN at low salinity, allowing KL2-R2 to occupy some of the available volume in the PN to form condensates.):

Why did the stable structure (KL2-R2) have a weaker interaction with the PN matrix? What is the mechanism of this phenomenon?

Response: As we also answered to your comment #3, KL1-R1 and KL2-R2 share the same stem sequence but have different tail sequences for binding fluorescent labels. As discussed above, we found that KL1-R1 indeed has a higher level of binding to the $p(A_{20-o})_n$ than KL2-R2. The origin of this binding interaction comes from the tail of the KL structure, i.e., the R1 sequence. We have added this new result in Supplementary Figure 16.

Supplementary Figure 16. Binding interaction between A_{20-o} , KL1-R1, and KL2-R2.

a, Simulated concentration distribution of complex sequences in a pool of A_{20-o} (400 μM), KL1-R1 (200 μM), and KL2-R2 (200 μM) at 30 °C based on rna06 model. Salt: 1 M Na^+ . **b**, Predicted minimum of free energy (MFE) proxy structure at 30 °C of A_{20-o} + KL1-R1 and A_{20-o} + KL2-R2. Tails of KL1-R1 and KL2-R2 are highlighted in orange. The binding between A_{20-o} and KLs are indicated by arrows.

We added additional discussion in the text:

NUPACK simulations further indicate that the origin of the preferred interaction between KL1-R1 and PN matrix is rooted in the tail structure (Supplementary Fig. 16).

6) Fig. 5e and p.12, l.356-360: Why did KL2-R2 form multiple condensates in the PN? This behavior is different from KL1-R1. Can the authors explain this phenomenon based on the surface tension difference in both RNA condensates? Or is there another hypothesis?

Response: Thank you for your very inspiring question. The suggestion from the surface tension perspective is very insightful and helpful for the interpretation of the data presented in Fig. 5e. We have deeply searched in the relevant literature in biomolecular condensate research and found similar structure explained from surface tension perspective^{6,7}. And we found that such mechanism is also applicable to our system:

As KL2-R2 has a higher melting point, it means that the binding strength among KL2-R2 in condensates is stronger and therefore KL2-R2 condensates possess higher surface tension than KL1-R1 condensates. Besides, as we have newly determined in the Supplementary Fig. 16 (See also our answers to your questions #3 and #5), KL1-R1 has stronger binding interaction with the

PN matrix than KL2-R2. In a ternary system of PN, KL1-R1, and KL2-R2, the surface tension (γ) will follow the relationship: $\gamma_{\text{PN/KL2-R2}} > \gamma_{\text{PN/KL1-R1}}$, so that KL2-R2 condensate will always be enveloped by KL1-R1 condensate to minimize its contact with the PN matrix and the whole system can minimize the free energy. There is an additional requirement that $\gamma_{\text{KL1-R1/KL2-R2}}$ should not be too high, which is applicable to our system, as KL1-R1 and KL2-R2 have very similar sequences and structures.

We added this discussion in the revised manuscript and cited the relevant papers:

This also helps to explain the overall architecture of the multiphase structures in Figure 5b, e from the perspective of surface tension (γ) of the ternary system of PN, KL1-R1, and KL2-R2^{6,7}. Since KL2-R2 has a higher T_m regarding the KL interactions, the binding strength among KL2-R2 in condensates is stronger and therefore KL2-R2 condensates possess higher surface tension than KL1-R1 condensates. In addition, KL1-R1 has a stronger binding interaction with the PN matrix than KL2-R2. This means $\gamma_{\text{PN/KL2-R2}} > \gamma_{\text{PN/KL1-R1}}$ applies, so that KL2-R2 condensate are preferably enveloped by KL1-R1 or expelled from PN to minimize its contact with the PN matrix.

7) Fig. 5: Are the asymmetric behaviors in the interaction of KL1-R1/KL2-R2 with PN related to not only the stability of KL1-R1/KL2-R2 condensates but also the kinetics in the formation process of both the condensates?

Response: Thank you for your inspiring question regarding the kinetics aspects. To investigate the influence of transcription kinetics on the asymmetric behavior of KL1-R1/KL2-R2 with PN matrix, we compared in-solution transcription kinetics of KL1-R1 and KL2-R2 to assess the sequence effect on the transcription efficiency. To quantify the transcribed KVs, we employed a strand displacement reaction to react with the dangling strands (R1 reporter with R1 in KL1-R1, R2 reporter with R2 in KL2-R2 in Supplementary Fig. 13a; see sequence in revised Supplementary Table 2). Overall, both structures show a similar transcription efficiency with a slight trend for a faster transcription of KL2-R2. However, since orthogonal KV transcription in PN favors a co-condensate of KL1-R1 with PN matrix (Fig. 5b), these results collectively underpin that KV-PN interaction are the dominant cause for co-condensate formation, and not faster transcription kinetics.

Supplementary Figure 13. Transcription kinetics of KL1-R1 and KL2-R2 in solution.

a, Scheme showing transcription of KL1-R1 (top) or KL2-R2 (down) in solution characterized by plate reader. To quantify the transcribed Kls RNA, dsDNA reporters with fluorophore-quencher pairs are present in solutions to react with dangling strand of transcribed Kls (R1 Reporter for KL1-R1, R2 Reporter for KL2-R2) by SDR, generating fluorescent signals. **b**, Transcription kinetics for KL1-R1 and KL2-R2 in solution with free promoter oligonucleotide, monitored by plate reader ($[NTP] : [Rx \text{ Reporter}] : [T_{KLx-Rx}] : [p] = 100 : 10 : 1 : 1$, 30°C , 6 mM Mg^{2+} , $2.5 \text{ U}/\mu\text{L T7 RNAP}$, $x = 1 \text{ or } 2$). $N = 2$ independent experiments. Error areas represent standard deviation.

We added experiment and discussion in the revised manuscript:

To ensure that this bias in structural development does not arise solely from different transcription efficiency, we conducted three control experiments: First, we compared the in-solution transcription efficiency of KL1-R1 and KL2-R2, finding out comparable transcription efficiency of both KL2-R2 and KL1-R1 (Supplementary Fig. 13).

8) Fig. 5: Does the asymmetric behaviors change if the ratio of the concentrations of the KL1-R1/KL2-R2 template DNAs in the PN?

Response: Thank you for your question. In addition to the experiment presented in the main MS with an equal template ratio of KL1-R1 to KL2-R2, we have now conducted in-PN Kls transcription experiments with altered templates ratios: (1) KL1-R1/KL2-R2 = 1/3 and (2) KL1-R1/KL2-R2 = 3/1 to investigate the transcription speed on the structure of the formed condensate at 30 mM Mg^{2+} . We added this new result in Supplementary Fig. 14.

Most importantly, even when the KL2-R2 is present in 3-fold excess (KL1-R1/KL2-R2 = 1/3), the result is qualitatively similar to equal stoichiometry (KL1-R1/KL2-R2 = 1/1, present in the MS). Hence, even when favoring KL2-R2 transcription (top row), the preferred interactions of KL1-R1 with the PN matrix dominate the structure formation. KL2-R2 condensates form on the surface of the PN. As a control (bottom row), we also inverted the ratio to have a higher concentration of KL1-R1 template inside the PN (KL1-R1/KL2-R2 = 3/1), where only KL1-R1 condensates can be found. Overall, these results suggest that KL-to-PN interactions, rather than transcription speed, dominates the co-condensate formation.

Supplementary Figure 14. Effect of asymmetric template concentrations for orthogonal KLs on the formation of condensates in PN.

Representative CLSM images of orthogonal KLs transcription in PN with 1/3 ratio (top) and 3/1 ratio (bottom) of T_{KL1-R1}/T_{KL2-R2} at 30 mM Mg^{2+} ([NTP] : [R1*] : [R2*] : [T_{KL1-R1}] : [T_{KL2-R2}] : [ρ] = 3.6 : 1.8 : 1.8 : 0.75 : 0.25 : 1, 30 °C, 2.5 U/ μ L T7 RNAP). Green channel: KL1-R1; Magenta channel: KL2-R2. Scale bar: 10 μ m.

We added the experiment and relative discussion in the revised manuscript:

Second, we further screened whether biasing the in-PN transcription towards KL2-R2 by increasing its template concentration to a 3-fold excess over the KL1-R1 template would change the structure formation. However, even under such conditions, the co-condensate morphology does not invert, but KL1 condensates remain inside the PN and KL2 condensates form on the surface of the PN (Supplementary Fig. 14). Taken together, these results provide mutually reinforcing evidence supporting that the interaction between different KL and PN dominates the formation of the co-condensates and not the pure transcription efficiency.

9) Fig. 5: If only the KL1-R1 condensate is degraded after the formation of Fig. 5b structures, does the KL2-R2 condensate enter the PN and re-form a single condensate in the PN like Supplementary Fig. 9b?

Response: Thank you for the very interesting question. Since it is not feasible to specifically degrade KL1-R1 without degrading KL2-R2, we provide an additional experiment based on some feasible assumptions to address this issue. We believe the question actually focuses on whether KL2-R2, existing outside of PN, would enter the pristine PN and form a co-condensate solely through KL-PN and KL interactions.

To investigate whether the KL-PN and KL interactions are strong enough to recruit KL2-R2 from outside of PN to form co-condensate, KL2-R2 was first transcribed in solution with $3.6 \times$ NTP for 18 h. This low concentration does not enable the formation of KL condensate in solution (Fig. 2e).

Then, PN was added and incubated with existing KL2-R2 for another 18 h at 30 °C with shaking (Supplementary Fig. 12c; NEW). The KL2-R2 transcripts are recruited into the PN, however, the structure is clearly different from co-transcriptional condensation that shows a spongy phase segregation (original Supplementary Fig. 9b, now in Supplementary Fig. 12b). Here, KL2-R2 recruitment into PN does occur, but no evident condensation/shrinkage of the structure occurs. This means that interactions between KL2-R2 and PN are able to partition some amount of transcript into the PN, but KL2/KL2 interactions are not abundant enough to initiate the same level of co-condensation as for in-PN transcription. This new result has now been added to Supplementary Fig. 12c.

Supplementary Figure 12. Formation of KL1-R1 and KL2-R2 condensates in PNs versus KL2-R2 transcribed in solution and recruited into PN.

a, Scheme and representative CLSM image showing the formation of single condensates in PN by localized transcription of KL1-R1. The condensate is labeled by R1*-Atto647 (green channel). **b**, Scheme and representative CLSM image showing the formation of single condensates in PN by localized transcription of KL2-R2. The condensate is labeled by R2*-Atto488 (magenta channel). 2.5 U/μL T7 RNAP, 30 mM Mg²⁺, 30 °C, [NTP] : [R1* or R2*] : [T_{KL1-R1} or T_{KL1-R1}] : [p] = 3.6 : 3.6 : 1 : 1, 18 h reaction for both (a) and (b). **c**, Scheme and representative CLSM images for the transcription of KL2-R2 in solution, which gets recruited into the PN. KL2-R2 was first transcribed in solution with UTP-Atto488 feeding ([NTP] : [T_{KL2-R2}] = 3.6 : 1, 1 mol% UTP-Atto488, 30 °C, 2.5 U/μL T7 RNAP, 30 mM Mg²⁺). Pristine PN was added and incubated with the existing KL2-R2 for 18 h for recruitment at 30 °C with 30 mM Mg²⁺. Green channel: PN shell labeled with o*-Atto647; Magenta channel: KL2-R2-Atto488. Scale bars are 5 μm for (a) and (b), and 10 μm for (c).

We added the experiment and relative discussion in the revised manuscript:

Third, we also probed whether KL2-R2 transcribed in solution can be enriched into pristine PN to check for interactions. Here, distinct differences are visible when comparing the structures formed by in-PN transcription of KL2-R2 versus KL2-R2 transcripts recruited from solution (Supplementary Fig. 12c). The in-PN transcription clearly induces phase segregation by KL2/KL2 interactions, whereas the latter rather points to some KL2-R2/PN interactions that enable a certain level of recruitment. Clear phase segregation via contraction of a spongy co-condensate phase is not

visible for the latter. Overall, the last control also emphasizes the critical role of in-PN transcription to facilitate co-condensate formation.

Reviewer #2 (Remarks to the Author):

I have read with great interest the manuscript by Xie, Chen et al. The report combines the DNA based protocells developed by the Walther group with recently introduced RNA nanostructures capable of forming condensates upon in vitro transcription. The team uses these technologies to study co-transcriptional condensation in the crowded microenvironments of the DNA protocells that, arguably, shares many similarities with condensate formation in eukaryotic nuclei.

The authors perform a wide array of experimental assays and controls and observe a number of intriguing effects including the key role played by interactions between the RNA nanostars and the surrounding DNA matrix in determining the characteristics (and presence) of the condensates.

Overall, I find this manuscript well written, and the results relevant to the interdisciplinary readership of the journal, particularly those interested in biomolecular condensates, synthetic cells and DNA/RNA nanotechnology. The document is well written and clear, and the figures are of high quality. I am happy to recommend publications after the authors have addressed the following minor comments:

Thank you for your very positive evaluation of our work!

1) In page 5, line 151, the authors introduce [NTP] as “defined as the maximum amount of KL1 transcripts that can be produced per template”. Could the authors clarify better this definition? Is this the overall NTP concentration divided by the number of nucleotides in one nanostar construct?

Response: Thank you for the comment. The [NTP] is calculated by the maximum amount of transcript [Transcript] divided by template concentration in the solution. [Transcript] is calculated by the concentration of the corresponding monomers divided by the number of the most abundant nucleotide in the transcript. For example, if U is the most abundant nucleotide in the transcript, we will calculate the [transcript] by taking the concentration of UTP divided by the number of U in transcript. To improve clarity, we have now added a new method section “[NTP] calculation” and refereed it in page 5 line 154.

Added text in method section:

[NTP] calculation

[NTP] is defined as the maximum amount of transcripts that can be produced per template given the nucleotide concentrations in the NTP mix in relation to the sequence of the transcript. For example for [NTP] : [template] = 5 : 1, the [NTP] concentration is set in a way to at least allow for 5 full transcripts from 1 template based on the most abundant nucleotide in the transcript. Other NTPs will be in a slight excess as the NTP mix has equal stoichiometry for all four needed NTPs.

2) In page 5, line 158, the authors state “This comparison demonstrates that the spatial transcription of the KL1 in PN leads to locally high concentrations sufficient for condensation, similar to the enrichment mechanism in natural nuclear condensates”, referring to the observation that under the relevant conditions KL1 condensates form in PN but not in bulk. However, we find out later in the manuscript that the effect is likely due to affinity between the nanostars and poly(A20-o), so this statement is not really correct. Perhaps the authors could simply refer the reader to the later parts of the manuscript for a rationalisation of this effect?

Response: Thank you for your very helpful suggestion.

To improve the clarity of the manuscript, we have added the following:

In addition, as we will demonstrate below, the KL-PN affinity also plays an important role in the condensation process.

3) Similarly, when discussing the difference in FRAP recovery between PN and bulk (page 5, line 166), the authors could refer to the subsequent discussion on interactions between the KL1 nanostars and the DNA matrix, which is probably behind the slower recovery seen in PN.

Response: Thank you for your helpful suggestion. We had mentioned this in the original manuscript in page 5, line 170-173, and now in page 6, line 174-177 in the revised manuscript, which might have slipped your eyes.

Text in original manuscript: “In contrast, half-bleached KL1 condensates in PN show less recovery and lack the bright edge, likely due to their restricted dynamics in a DNA-crowded environment and interactions between PN matrix and the KL1 transcripts, as we will further discuss below.”

4) In Figs 2 c and k the authors convert a fluorescent signal from a molecular beacon into RNA concentrations. Could they please provide information on how this is done?

Response: Thank you for your question. We assume that the transcribed RNA concentration is equal to the concentration of the activated reporter, which is reflected by the measured fluorescence intensity. And in parallel, we have a pure DNA-fluorophore conjugate sample at 1000 nM, which serves as reference. By calculating the intensity ratio between the individual samples and the reference, we can obtain the RNA concentration in individual samples. We have now included a more detailed description for calculating RNA concentration in the Method section in revised manuscript.

For each plate reader experiment, we have included a reference sample containing 1000 nM DNA-fluorophore conjugate. We calculate the intensity ratio between individual samples and the reference sample to yield the transcribed RNA concentration by following equation:

$$[\text{RNA}] = 1000 \text{ nM} \times \frac{\text{Intensity (sample)}}{\text{Intensity (reference)}}$$

5) The authors use a single-stranded DNA template for transcribing the RNA nanostars. This will probably fold into a DNA nanostar similar to the RNA version, which may hinder transcription. Have the authors tested transcription using a double-stranded template? Does this change transcription efficiency in PN or the bulk? Have they included GU wobble pairs in the RNA nanostar designs to destabilise the secondary structure of the single-stranded template?

Response: Thank you for your question. For KL (RNA nanostars) transcription, we **actually used dsDNA as template**, as we had the same concern that the ssDNA template might hinder the RNA transcription due to the internal hairpin structure. There is only use of ssDNA template for linear transcript in Figure 2 c, k. We did not add GU wobble pairs in the KL designs, as we used dsDNA, which avoids the hairpin structure formation inside the template. We have noticed that the scheme in the original Figure 1 was not clear enough. We have now modified the Figure 1 with a dsDNA template to improve the accuracy of the scheme.

In addition, in the revision, we tested whether ssDNA templates enable proper transcription. As expected, transcription is very inefficient. Condensate formation is absent after 24 h for both in-solution and in-PN transcription (Supplementary Fig. 2).

We added this result to the new Supplementary Fig. 2, and referenced it in the text:

As a proof of concept, we first focus on a three-armed singled-stranded RNA (ssRNA) nanostar with a wildtype palindromic KL sequence^{1,8,9} at the tip of each arm (KL1 in Fig. 2d). We used a dsDNA template (T_{KL1}/T_{KL1}') because ssDNA templates (T_{KL1}) alone do not allow for efficient transcription and condensate formation on account of intramolecular folding of such ssDNA template (Supplementary Fig. 2). T_{KL1} contains a p* ssDNA sequence for hybridization to poly(A₂₀)_n inside the PN to initiate transcription (sequences in Supplementary Table 2).

Supplementary Figure 2. ssDNA template hinders the formation of transcriptional KL condensates in both solution and PN.

a, Representative CLSM images of transcriptional KL condensate in solution with $[NTP] : [T_{KL1}] = 7.2$, transcribed by promoter ssDNA, which is hybridized with T_{KL1} as a ssDNA template or T_{KL1}/T_{KL1}' as a dsDNA template ($[T_{KL1}] : [p] = 1 : 1$, 30 °C, 30 mM Mg^{2+} , 2.5 U/ μ L T7 RNAP). Note that $[NTP]$ is set to a concentration where condensation in solution appears (see Supplementary Figure 3). **b**, Representative CLSM images of transcriptional KL in PN with ssDNA

template or dsDNA template for transcription ([NTP] : [T_{KL1}] = 3.6, 30 °C, 30 mM Mg²⁺, 2.5 U/μL T7 RNAP). Scale bar: 10 μm.

6) Following up from 5: the authors ascribe the preferential condensation of KL1 stars within PN to the affinity with the o domain in poly(A20-o), and they show good evidence for it. However, 10% of the scaffold is DNA in the PN poly(A20-p), hybridized to single-stranded DNA templates that are likely to bind the transcription products (particularly when transcription ends and the polymerase loses activity). The authors could perhaps test or discuss this possibility?

Response: Thank you for your question.

As our answer to your question #5, we actually used dsDNA templates for transcription of KL nanostars throughout the whole paper. Therefore, strong interaction between the transcribed products with the template cannot be the major cause.

Reviewer #3 (Remarks to the Author):

Within a core-shell DNA coacervates, the authors demonstrated that specific RNAs (kissing loop sequence, KLs) can be transcribed in-situ to form another phase of condensate inside, and this platform is termed as protonuclei (PN). To construct this, they designed polyA and polyT containing ssDNA by incorporating the T7 promoter sequence or other barcode DNAs, and mixed with T7 DNA ligase. They provided the mechanism of co-condensation of transcriptional KL1 with the PN DNA matrix. This platform is capable of transcribing two KLs simultaneously that produces different patterns of multiphasic condensates by changes in magnesium concentration.

This work is original because the authors provided the first demonstration of in-situ co-transcription condensation KLs in PN that can form multiphasic condensates. This work provides a clear observation of the KLs transcribed in PN to undergo condensation with DNA matrix. This work provides the new design strategy to construct the synthetic system that can perform the in-situ transcription with modality. This is potentially an interesting system for artificial cell research and synthetic biology.

Thank you for your very positive evaluation of our work!

While they provide thorough experimental designs to support their claims, there are a couple of remaining questions regarding their interpretation and observations:

1. If this core-shell DNA coacervates have a highly concentrated DNA-enriched core, why does the KLs transcription start from the inner-interface of PN shell (Fig 3b)? Is this a combined effect of NTPs diffusion and T7 polymerase from outside of DNA PN? Perhaps, the T7 polymerase is enriched at the shell?

Response: Thank you for your comment. We agree that the phenomenon of the initial transcription at the inner interface of PN shell is a result of the diffusion of both NTPs and T7 RNAP.

Although NTPs are relatively small molecules (~ 500 Da), they need to be transported from outside to the PN interior, and they get continuously consumed by T7 RNAP during transcription. Therefore, the outer part has the first access to the NTPs and can have higher transcription rate of KFs there. Besides, T7 RNAP is relatively large, with a molecular weight of around 100 kDa with slower diffusion. As the T7 RNAP gets into the PN through diffusion and binding to the template, the inner interface of the PN shell will have the first interaction and binding with the T7 RNAP. Therefore, the T7 RNAP will get firstly enriched at that part and induces transcription from there at the beginning of the process, which also leads to a local high transcription rate of KFs. Over time, this transcription kinetics difference throughout the PN will disappear, as the T7 RNAP become homogenous inside the PN by diffusion.

To confirm our hypothesis, we performed KL1 transcription in PN pre-equilibrated with T7 RNAP to eliminate the effect of T7 RNAP diffusion. To do so, we charged the dsDNA template and T7 RNAP into PN 2 h prior to NTP addition. Upon the NTP addition to initiate transcription, KL1 intensity shows a more uniform increase in PN during the first 6 h transcription due to the continuous NTP consumption and the slow diffusion of NTP. This result implies that the diffusion of both T7 RNAP and NTP plays a role in the KL1 transcription in PN. The co-transcriptional phase segregation is however very similar. We have added this new result to Supplementary Figure 6, as also shown below.

Supplementary Figure 6. Pre-equilibrium of T7 RNAP enables a more homogeneous transcription inside PN.

Rationale: To investigate if the diffusion of T7 RNAP into PN has an impact on the peripheral transcription, we performed transcription of KL1 inside PN equilibrated with T7 RNAP prior the addition of NTPs. **a**, scheme and representative CLSM images of KL1 transcription in PN pre-equilibrated with T7 RNAP for 2 h, before the addition of NTPs to trigger transcription. The plot shows cross-sectional line profile along the white line in the CLSM image at 6 h, demonstrating a more homogeneous transcription inside the PN. **b**, scheme and representative CLSM images of KL1 transcription in PN with simultaneous addition of T7 RNAP and NTPs (standard conditions used in the main text). The plot shows cross-sectional line profile along the white line in the CLSM image at 6 h, demonstrating a non-homogeneous and peripheral transcription inside the PN. Green channel: KL1 condensate labeled by UTP-Atto488; Magenta channel: PN shell ($\text{poly}(\text{T}_{20}\text{-k})_n$ labeled with $\text{k}^*\text{-Atto647}$). $[\text{NTP}] : [\text{T}_{\text{KL1}}] = 3.6 : 1$, 1 mol% UTP-Atto488, 30 °C, 2.5 U/ μL T7 RNAP, 30 mM Mg^{2+} . Note that we only focus on the first 6h of co-transcriptional condensation, because afterwards it's rather the maturation of the structure/co-condensate. Scale bar: 10 μm .

We added new text to discuss this point in the revised manuscript:

In the first 12 h, transcription takes place from the edge of the PN to their center due to the continuous consumption of NTPs as well as diffusive uptake of T7 RNAP and NTPs. The KL1 intensity gradient can be diminished if the PN are pre-equilibrated with T7 RNAP (for 2 h) prior to the addition of the NTPs (Supplementary Fig. 6). The overall structure formation proceeds however in a very similar fashion.

2. In this context, the authors describe the slower rate by T7 RNA polymerase in PN caused by the slow diffusion of NTPs. Supposedly, this highly concentrated ssDNAs at the interface could suggest more dense networks of DNAs. Then, I would expect that T7 RNA polymerase diffusion would be slow as well, or even limited, because it is bigger than the NTPs. If this transcription is in a kinetic regime where the enzyme concentration doesn't affect the overall rate of reaction, the authors point of view is valid. Otherwise, some discussion regarding the T7 RNA polymerase encapsulation in PN needs to be addressed.

Response: Thank you for your insightful and inspiring question. You are right that the T7 RNAP has much higher molecular weight (~ 100 kDa) than NTPs (~ 500 Da), so that the diffusion of T7 RNAP could potentially also influence the transcription inside the PN. Indeed, for the result shown in Fig. 2c in the original manuscript, the experiment was performed by directly adding T7 RNAP and NTPs into the solution to trigger the transcription. In this case, T7 RNAP needs certain time to diffuse into the PN and get encapsulated, which likely hinders the transcription kinetics. According to our new result shown in Supplementary Fig. 6 (for your comment just above), pre-equilibrating PN with T7 RNAP diminishes the gradient intensity of KL1 during its transcription, which confirms the encapsulation of T7 RNAP within PN (see detailed answer to your question #1). In this case, different concentrations of T7 RNAP could affect its diffusion rate into PN, potentially influencing transcription kinetics.

To investigate whether the diffusion of T7 RNAP with applied concentration in the paper (2 U/ μ L or 2.5 U/ μ L) affects the overall transcription kinetics, we now performed new experiments by using various concentrations of T7 RNAP (ranging from 2-10 U/ μ L) without pre-incubation. The result shows that the RNA polymerase concentration has a slight influence on the transcription rate. We just mention these results here to not clutter the manuscript and the SI. According to the transparent review process, this information will still be available for the reader.

Figure for Review 1. In-PN transcription kinetics with various T7 RNAP concentrations.

Higher concentrations of T7 RNAP facilitate faster diffusion and encapsulation into PN, thereby enhancing transcription kinetics. The experiment was monitored by SDR of the R in a plate reader ($[NTP] : [R] : [T_{Rep^+}] : [p] = 2000 : 10 : 1 : 1$, 30 $^{\circ}$ C, 6 mM Mg^{2+} , 2 – 10 U/ μ L T7 RNAP). $N = 3$. Shaded areas represent the standard deviation. Note

that the transcription kinetics differ from the data shown in Fig. 2c in the original manuscript due to different enzymatic activity from different batches.

We also modified the main text to discuss the influence of diffusion of T7 RNAP on the slower transcription kinetics in PN:

The slightly lower activity can be understood considering constraints of the diffusion of T7 RNAP and NTPs into the PN, and RNA strands out of the PN.

3. Interpretation about the dsDNA exclusion from KL1 condensates in Figure 4 needs clarification.

a. If KL1 sequence does not have any specific interactions with the PN matrix, why would the hybridization of ssDNA with o* make KL excluded from the condensates? What is this interaction between KL1 and PN DNA matrix? Nonspecific interactions between KL and PN DNA matrix could be the charge interaction mediated by magnesium. Complementary sequence for PN DNA matrix would likely have partial dsDNA structure, which can yield in the increased charge density overall. If so, wouldn't it make these nonspecific interactions between KL and the polyA matrix stronger? Perhaps, magnesium is depleted to mediate the nonspecific charge interactions between PN matrix and KL1 that leads to the exclusion of the KL1? This could partially explain why the condensates inside of PN become dominant by DNA matrix. I wonder higher magnesium concentration will make the co-condensates to be persisted upon adding o* invader.

Response: We believe that KL (no R1/R2 tail) and PN have non-specific binding interaction, which comes from the hydrogen-bond between individual nucleobases. The Mg^{2+} also induces some extent of charge interaction for the binding between phosphate backbones. However, when hybridizing the o* in the PN matrix, the overall net negative charge density largely increases, which assists to expel the KL1, which is also negatively charged, due to electrostatic repulsion. Besides, hybridization also increases the persistence length of matrix strands in PN. This enhances the steric repulsion between PN matrix and the KL1. Therefore, the exclusion of the KL1 is a result of both electrostatic repulsion and steric repulsion, induced by o/o* hybridization.

High magnesium concentration helps to stabilize the co-condensates, by screening the electrostatic repulsion. To verify this, we have now performed new experiments by firstly forming KL-PN co-condensates at 30 mM Mg^{2+} and then increasing the overall Mg^{2+} in solution to 100 mM, before adding o*-Atto647 invader strand. While KL-PN co-condensates at 30 mM Mg^{2+} is completely disassembled 1 h after o*-Atto647 addition (Fig. 4c), the co-condensate still exists after 2 h of o*-Atto647 invasion (Supplementary Fig. 11). After 24 h of o*-Atto647 addition, PN-KL co-condensate is disassembled, resulting in the occupation of poly(A₂₀-o)_n/o*-Atto647 in the majority of PN, with very few KLS remaining inside PN. This result implies that a higher Mg^{2+} concentration indeed helps stabilize KL interactions as well as the PN-KL co-condensates, thereby slowing down the disassembly of the co-condensate. However, charge screening with a higher Mg^{2+} concentration could not completely preserve the integrity of KL-PN co-condensate.

Supplementary Figure 11. Invasion of o*-Atto647 at 100 mM Mg²⁺ shows slower disassembly of KL-PN co-condensate compared to 30 mM Mg²⁺.

a, Representative CLSM images of KL-PN co-condensate formed by KL1 transcription in PN with 30 mM Mg²⁺ (KL1-Atto488, green channel). The overall Mg²⁺ concentration was then increased to 100 mM, and the solution was incubated for 24 h before the addition of o*-Atto647 to hybridize with poly(A₂₀-o)_n in PN (magenta channel). **b**, Representative CLSM images of KL-PN co-condensate at 2 or 24 h after o*-Atto647 addition at 100 mM Mg²⁺. Scale bar: 10 μm.

We added new texts to discuss this result:

Moreover, an invasion process of o*-Atto647 at 100 mM Mg²⁺ shows a slower disassembly of the co-condensate compared to 30 mM Mg²⁺ (Fig. 4c), implying the critical role of Mg²⁺ in stabilizing interactions of KLS and KL-PN (Supplementary Fig. 11).

b. Why does the KL1 condensate morphology by the prehybridization of DNA matrix in Fig 4f differ from Fig 4a? Is it because of different time point of imaging? Maybe the morphology in Fig 4a is kinetically arrested?

Response: Thank you for your question. This question is similar to reviewer 1 - question #2. In Fig. 4a and c, RNA condensates are first formed in PN, which are **NOT** hybridized with o*-Atto647 (o*-Atto647 has the function to reduce affinity between KL condensates and the PN interior). o*-Atto647 is later added as an invader after RNA condensates were formed. That is why a spherical RNA condensate gradually dissolves from the outer surface to the center of PN.

In contrast, in Fig. 4f, KL transcription and condensation is conducted in PN **prehybridized** with different amounts of Atto647-o*. This hybridization weakens the KL-to-PN affinity, making RNA condensation within the PN difficult at 10% prehybridization and leading to ill-defined RNA condensates. One can also see the brighter edges of these condensates. These arise from rather pure KL condensates, whereas the less intense core is a co-condensate. The pure KL layer even

appears to dewet from the central condensate as seen by the undulations/non-perfect bright shell. At higher prehybridization levels, the internal PN environment becomes more repulsive, which prevents RNA condensation inside PN. Therefore, small condensates form at the outer shell of PN due to diffusion of KL transcripts from the PN to the surrounding.

We find that the experimental description is insufficient in the original manuscript for showing the differences between experimental settings for Fig. 4a-c and Fig. 4f. Therefore, we now provide additional text and explanation for Fig. 4a-c as follows:

To do so, we added different stoichiometric amounts of o*-Atto647 into solutions of PN containing already formed KL1-poly(A₂₀-o)_n co-condensates and investigated how the invasion by o*-Atto647 would alter the pre-existing KL1-poly(A₂₀-o)_n co-condensates. A gradual invasion of o*-Atto647 into the KL1-poly(A₂₀-o)_n co-condensates takes place as the amount of o*-Atto647 increases (Fig. 4a, b). A continuous surface erosion of the co-condensates occurs because the o/o*-Atto647 hybridization reduces the affinity between KL condensate and PN interior by introducing stronger electrostatic and steric repulsion inside PN due to increased negative charge density and increasing persistence length of the formed dsDNA parts^{4,5} (Fig. 4c-e and Supplementary Video 3).

For Fig. 4f, we have also modified the original text and believe the relevant discussion (highlighted) now can stand alone and make a clear distinction to the discussion about Fig. 4a-c:

“Seeing such a profound impact, we then investigated KL1 transcription in PN with a poly(A₂₀-o)_n matrix pre-hybridized by different amounts of o*-Atto647 (from 0% – 300%) to provide weakened affinity between PN matrix and KL1 transcripts. In analogy with the above result, single KL1 condensates form in pristine PN (Fig. 4f). When applying 10% o*-Atto647, the KL1 transcripts form single condensates with irregular secondary nucleation on its surface inside the PN, along with multiple tiny nuclei outside the PN shell (Fig. 4f). The brighter green parts are condensates purely enriched with KL1 transcripts that remain inside the PN due to relatively sufficient affinity. The marked difference – heterogeneously structured RNA condensates in Fig. 4f compared to the rather spherical and homogenous structures formed after invasion of pre-formed condensate in Fig. 4a, c – can be attributed to their different formation processes. Increasing the content of pre-hybridized o*-Atto647 domains from 10% to 300% gradually prevents KL1 condensate formation inside the PN due to weakened PN-KL1 interaction, which likely becomes even repulsive at higher pre-hybridization degrees because the ssDNA to dsDNA transition leads to higher negative charge density and persistence length, allowing for stronger electrostatic and steric repulsion^{4,5}, respectively, inside the PN. As a result, the KL1 transcripts formed inside the PN do not yield condensates inside the PN, instead, multiple small transcriptional KL1 condensates form in the PN surroundings.”

c. dsDNA can have greater persistence length that leads to exclusion from protein membraneless organelles in-vitro (Nott, T., Craggs, T. & Baldwin, A. Membraneless organelles can melt nucleic acid duplexes and act as biomolecular filters. Nature Chem 8, 569–575 (2016), some discussion

about dsDNA vs ssDNA in Jeffrey R. Vieregge et al, Journal of the American Chemical Society 2018 140 (5), 1632-1638). I wonder whether these dsDNA material properties affect its partitioning trends in Fig 4 g & h.

Response: Thank you for the insightful suggestion. Yes, indeed, the greater persistence length of dsDNA is relevant for interpreting the partitioning trends in Fig. 4g, h. As reported in the suggested two literatures, dsDNA has higher charge density and is less flexible (*JACS*, **140**, 1632-1638 (2018)), and the different properties of dsDNA and ssDNA have been shown to affect their partition into the membraneless organelles (*Nat. Chem.* **8**, 569–575 (2016)). We believe that these are important references to demonstrate the two critical properties of dsDNA, i.e., higher charge density and greater persistence length for stronger electrostatic repulsion and steric repulsion, which helps to explain the partitioning trend observed in Fig. 4g, h, that dsDNA is excluded while ssDNA is enriched in the transcriptional KL condensates in solution. In addition, we believe the persistence length aspect of dsDNA also plays an important role in the invasion experiments shown in Fig. 4a-f.

We have now cited the referred papers and added relevant texts to discuss the persistence length and steric repulsion of dsDNA for the relevant results:

Increasing the content of pre-hybridized o*-Atto647 domains from 10% to 300% gradually prevents KL1 condensate formation inside the PN due to weakened PN-KL1 interaction, which likely becomes even repulsive at higher pre-hybridization degrees because the ssDNA to dsDNA transition leads to higher negative charge density and persistence length, allowing for stronger electrostatic and steric repulsion^{4,5}, respectively, inside the PN.

Furthermore, electrostatic repulsion from increased negative charge density, and steric repulsion from higher persistence length after dsDNA formation also play important roles^{4,5}, as in analogy to re-entrant phenomena in living cells.

4. It is still unclear how these two different KLs localize at the different locations under different magnesium concentration (Fig 5).

a. KL2-R2 condensates still have little amount of KL1 in both cases of magnesium concentrations, suggesting that the coexistence of KL1-R1 condensate and KL2-R2 condensate is more than sum of each individual condensates. Likely, DNA matrix for PN is involved. Please address that.

Response: Thank you for your interesting question. In Fig. 5a-d, for the two KLs transcriptions at 30 mM Mg²⁺, the budding KL2-R2 condensates do not contain PN matrix, as the PN matrix is trapped within the PN shell. Therefore, the PN matrix is not involved in the case of 30 mM Mg²⁺. While it is true that we observe the existing intensity of KL1-R1 at the KL2-R2 condensate. But the intensity is lower than KL1-R1 at the PN shell. We assume that the remaining intensity is due to the very few KL1-R1 arrested by slow diffusion kinetics at the KL2-R2 condensate.

In the case of 15 mM Mg²⁺ transcription, there is indeed a minor distribution of KL1-R1 in KL2-R2, or KL2-R2 in KL1-R1 (Fig. 5e, f). Both KL1-R1 and KL2-R2 are transcribed in PN, where KL1-R1 or KL2-R2 phase separates individually. The remaining intensity is likely due to the slow diffusion of the transcripts in the crowded environments.

We added text to explain this:

Note that a minor distribution of KL1-R1 in KL2-R2, or KL2-R2 in KL1-R1 (Fig. 5e, f), can be observed, possibly due to the slow diffusion of the produced KLS in the crowded condensates.

b. The reasoning of different morphology of KL condensates based on its material properties seem valid. This is a minor question, but do KL2 get transcribed inside of PN and get transported to be outside of shell? Or does the KL2 get transcribed outside of the DNA PN at 30 mM Mg²⁺?

Response: As we show in Supplementary Fig. 1, promoter sequences on poly(A_{20-p})_n are enclosed in the PN after phase separation, so that the KL2-R2 is transcribed inside PN and transported to the outside later by the repulsion from KL1-R1.

5. In line 428, what does it mean by helper proteins?

Response: Thank you for your question. The helper proteins are proteins that are involved in transcription elongation and RNA processing, such as P-TEFb as discussed for instance in literature¹⁰.

We modified the text in the revised manuscript:

While we focus on a rather artificial and well controllable system of KL condensates, this work lays an important cornerstone to study more sophisticated phase separation processes, such as in case of polymerase II that forms rich condensate architectures with helper proteins, e.g., P-TEFb that are involved in transcription elongation, and those which are implicated in disease and ageing^{10,11}.

Additionally, the manuscript does not specify the number of independent replicates. Some of fluorescence intensity curves have shaded areas representing standard deviations, but it is unclear where this standard deviation is coming from. Clarifying this information is important to improve the transparency and reproducibility of the study. The details of methods seem enough.

Response: Thank you for your comment. We have now included the number of independent repeats in the figure captions in revised manuscript and supporting information.

References

1. Fabrini, G. et al. Co-transcriptional production of programmable RNA condensates and synthetic organelles. *Nat. Nanotechnol.* **19**, 1665-1673 (2024).

2. Morita, M. et al. Liquid DNA Coacervates form Porous Capsular Hydrogels via Viscoelastic Phase Separation on Microdroplet Interface. *Adv. Mater. Interfaces* **11** (2024).
3. Liu, W., Lupfer, C., Samanta, A., Sarkar, A. & Walther, A. Switchable Hydrophobic Pockets in DNA Protocells Enhance Chemical Conversion. *J. Am. Chem. Soc.* **145**, 7090-7094 (2023).
4. Nott, T.J., Craggs, T.D. & Baldwin, A.J. Membraneless organelles can melt nucleic acid duplexes and act as biomolecular filters. *Nat. Chem.* **8**, 569-575 (2016).
5. Viereggs, J.R. et al. Oligonucleotide-Peptide Complexes: Phase Control by Hybridization. *J. Am. Chem. Soc.* **140**, 1632-1638 (2018).
6. Feric, M. et al. Coexisting Liquid Phases Underlie Nucleolar Subcompartments. *Cell* **165**, 1686-1697 (2016).
7. Shin, Y. & Brangwynne, C.P. Liquid phase condensation in cell physiology and disease. *Science* **357**, eaaf4382 (2017).
8. Stewart, J.M. et al. Modular RNA motifs for orthogonal phase separated compartments. *Nat. Commun.* **15**, 6244 (2024).
9. Udono, H. et al. Programmable Computational RNA Droplets Assembled via Kissing-Loop Interaction. *Acs Nano* **18**, 15477-15486 (2024).
10. Changiarath, A. et al. Promoter and Gene-Body RNA-Polymerase II co-exist in partial demixed condensates. Preprint at 10.1101/2024.03.16.585180 (2024).
11. Pei, G.F., Lyons, H., Li, P.L. & Sabari, B.R. Transcription regulation by biomolecular condensates. *Nat. Rev. Mol. Cell Bio.* **26**, 213–236 (2025).